# Design, Synthesis, and Development of pyrazolo[1,5-*a*]pyrimidine Derivatives as a Novel Series of Selective PI3K*δ* Inhibitors: Part I—Indole Derivatives

**DOI:** 10.3390/ph15080949

**Published:** 2022-07-30

**Authors:** Mariola Stypik, Marcin Zagozda, Stanisław Michałek, Barbara Dymek, Daria Zdżalik-Bielecka, Maciej Dziachan, Nina Orłowska, Paweł Gunerka, Paweł Turowski, Joanna Hucz-Kalitowska, Aleksandra Stańczak, Paulina Stańczak, Krzysztof Mulewski, Damian Smuga, Filip Stefaniak, Lidia Gurba-Bryśkiewicz, Arkadiusz Leniak, Zbigniew Ochal, Mateusz Mach, Karolina Dzwonek, Monika Lamparska-Przybysz, Krzysztof Dubiel, Maciej Wieczorek

**Affiliations:** 1Celon Pharma S.A., ul. Marymoncka 15, 05-152 KazuńNowy, Poland; marcin.zagozda@celonpharma.com (M.Z.); stanislaw.michalek@celonpharma.com (S.M.); bdymek@op.pl (B.D.); dzdzalik@iimcb.gov.pl (D.Z.-B.); maciejdziachan@gmail.com (M.D.); orlowska.nina@gmail.com (N.O.); pgunerka@gmail.com (P.G.); tupaw@wp.pl (P.T.); joanna.hucz@celonpharma.com (J.H.-K.); apstanczak@gmail.com (A.S.); paulinaseweryna.stanczak@gmail.com (P.S.); kmulewski91@gmail.com (K.M.); damian.smuga@celonpharma.com (D.S.); stefaniak@gmail.com (F.S.); lidia.gurba@celonpharma.com (L.G.-B.); arkadiusz.leniak@celonpharma.com (A.L.); mateusz.mach@celonpharma.com (M.M.); karolina.dzwonek@gmail.com (K.D.); lamparska@poczta.onet.pl (M.L.-P.); krzysztof.dubiel@celonpharma.com (K.D.); maciej.wieczorek@celonpharma.com (M.W.); 2Faculty of Chemistry, Warsaw University of Technology, ul. Noakowskiego 3, 00-664 Warsaw, Poland; ochal@pw.edu.pl

**Keywords:** PI3K*δ* inhibitors, Asthma, COPD, 5-indole-pyrazolo[1,5-*a*]pyrimidine, CPL302253

## Abstract

Phosphoinositide 3-kinase *δ* (PI3K*δ*), a member of the class I PI3K family, is an essential signaling biomolecule that regulates the differentiation, proliferation, migration, and survival of immune cells. The overactivity of this protein causes cellular dysfunctions in many human disorders, for example, inflammatory and autoimmune diseases, including asthma or chronic obstructive pulmonary disease (COPD). In this work, we designed and synthesized a new library of small-molecule inhibitors based on indol-4-yl-pyrazolo[1,5-*a*]pyrimidine with IC_50_ values in the low nanomolar range and high selectivity against the PI3K*δ* isoform. CPL302253 (**54**), the most potent compound of all the structures obtained, with IC_50_ = 2.8 nM, is a potential future candidate for clinical development as an inhaled drug to prevent asthma.

## 1. Introduction

PI3Ks (phosphoinositide 3-kinases) are a family of lipid kinases that can perform the phosphorylation reaction of the hydroxyl group at the 3-position of the phosphatidylinositol ring. More specifically, they are capable of catalyzing the phosphorylation reaction of 4,5-phosphatidylinositol diphosphate (PIP2) to 3,4,5-phosphatidylinositol triphosphate (PIP3) [1,2,3]. This family of kinases consists of three classes (I, II, and III) in terms of the structure and affinity for the substrate. Most class Is of PI3Ks have been described in the literature. PI3K I consist of heterodimeric proteins: PI3K*α*, PI3K*β*, PI3K*γ*, and PI3K*δ* [1,2,3,4]. Each of them is involved in different functions and cellular processes, such as proliferation, migration, cytokine production, or apoptosis [1,2,3,4]. Cells involved in the body’s immune response, such as macrophages, neutrophils, T, and B cells, highly expressed PI3K*γ* and PI3K*δ* [1,2,3,4,5]. The role of PI3K*δ* as the co-stimulator between T to B cell interactions was also reported [6,7]. In addition, two other subunits, PI3K*α* and PI3K*β*, are involved in normal embryogenesis or metabolism regulation. Therefore, PI3K*δ* has been identified as an attractive and promising therapeutic target for the treatment of cancer, autoimmune and inflammatory diseases [8,9,10,11,12,13,14].

One of the manifestations of inflammatory diseases is asthma, a chronic illness with a spectrum of respiratory symptoms burdensome for patients [15,16,17]. It was reported that PI3K*δ* is involved in the regulation of allergic asthma development processes, such as activation of cytokines expression by Th2 cells, activation of antibodies production (e.g., IgE) by B cells, activation of basophils, and accumulation following the migration of eosinophil in the lungs [2,15,18]. Thus far, several selective PI3K*δ* inhibitors have been developed, to name only: Idelalisib (PI3K*δ* selective) or Duvelisib (PI3K*δ* and *γ* selective; Figure 1) [15,19,20,21]. Unfortunately, the toxicity and side effects caused by these candidates’ low selectivity in systemic action exclude them from the group of potential future therapeutics for asthma management [15,22,23]. Therefore, new approaches focused on developing safe, selective PI3K*δ* inhibitors designed to be conveniently delivered by inhalation remain an unfulfilled challenge [15,23]. Rich expression of PI3K*δ* by lung epithelial cells provides the rationale for the new drug design against asthma as the alternative for patients poorly responding to current treatments.

The therapeutic application of PI3K*δ* inhibition at the molecular level utilizes particular interactions of the respective inhibitors within the p110*δ* subunit of the ATP binding site [24,25]. Several binding protein key sites are involved in this mechanism: the affinity pocket, the hinge pocket, and a hydrophobic region located below the non-conserved part of the enzyme’s active site [25,26,27]. Numerous active PI3K*δ* inhibitors are characterized by the interactions with a conserved tyrosine residue (Tyr-876) and hydrogen bonds with Lys-833 located at the binding pocket [27,28]. Most selective PI3K*δ* inhibitors, however, form a specific hydrogen bond between two critical amino acids: Trp-760 and Met-752 [24,28,29]. In addition, opening the pocket between the Trp-812 and Met-804 has been identified as a selectivity improvement operation [25]. Moreover, PI3K*δ* selectivity strongly depends on the interaction with Trp-760, for which a ‘tryptophan shelf’ term was coined [6,24,25]. Binding to Asp-787 was also observed.

Many inhibitors of PI3K have been designed and developed to date. Of the small molecules [12,30,31] and non-specific inhibitors (pan-PI3K) PI-103 [32], ZSTK474 [33], Pictilisib (GDC-0941) [34], Copanlisib (BAY80-6946) [35] and Buparlisib (BKM-120) can be mentioned [36]. More selective inhibitors for particular enzyme isoforms were later developed, such as, e.g., Apitolisib (GDC-0980) [37], Idelalisib (CAL-101) [38], and Duvelisib (IPI-145) were developed [39]. Most of them are applied in cancer therapies [31]. Only a few PI3K*δ* or PI3K *γ*/*δ* inhibitors have been considered potential drugs in the treatment of respiratory diseases such as chronic obstructive pulmonary disease (COPD) and asthma, namely Nemiralisib (GSK2269557) [40], RV-1729 [41], LAS195319 [42] and AZD8154 [43,44]. Among them, Nemiralisib (Figure 1, terminated in phase II clinical trials) [45] and GSK-2292767 (which did not cross phase I) were delivered by inhalation route [6,46]. In autoimmune and immunodeficiency diseases therapeutic area, two oral PI3K*δ* inhibitors have advanced to clinical phase three development: Leniosilib and Seletalisib [5,47,48,49].

Most of the pan-PI3K inhibitors hold in their molecular structure bicyclic cores such as thienopyrimidines (GDC-0941), purines, pyridopyrimidines, or furopyrimidines (Figure 1) [6,27]. The enormous activity and selectivity potential have been associated with the presence of the morpholine ring in the “morpholine-pyrimidine” system (marked in red in Figure 1) [6]. In the hinge-binding mechanism motif, the morpholine ring plays a role as an *H*-bond acceptor. The heteroaromatic or aromatic ring (marked in green in Figure 1), placed in a “meta”-like position to the morpholine ring, takes up space within the affinity pocket of the enzyme (binding to Val-828) [6,25,27]. This mutual interaction enhances the activity and selectivity of designed inhibitors. Moreover, the heterocyclic system (marked in blue in Figure 1) occupying the pocket responsible for the kinase’s specificity drives the selectivity of the designed compounds [6,25,27].

In our work, utilizing known “morpholine-pyrimidine” structure-PI3K*δ-*activity relationship and bicyclic pyrazolo[1,5-*a*]pyrimidine core, we developed a novel library of compounds focused on future COPD treatment. More specifically, we were fixed on the substitution of morpholine at the C(7) position leading to the 7-(morpholin-4-yl) pyrazolo[1,5-*a*]pyrimidine structural motif. According to mentioned in the above paragraphs’ correlations, we focused on the pyrazolo[1,5-*a*]pyrimidine core as probably the most promising structure (including the nitrogen atom in the five-membered ring), especially with the morpholine moiety in the appropriate position (to create the “morpholine-pyrimidine” system). We noticed that based on the structure of inhibitors as the candidates for the treatment of COPD or Asthma, cores based on bicyclic rings five-six-membered are more potent than six-six-membered, such as in CDZ 173 or UCB-5857. Moreover, we hoped that a five-six-membered ring, similar to pan-inhibitor GDC-0941 with appropriate modifications, could improve and increase the selectivity for isoform *δ* and thus becomes a selective PI3K*δ* inhibitor. As a result, we obtained a selection of indole derivatives with improved potency and selectivity towards PI3K*δ* inhibition. Moreover, we observed that 5-indole-pyrazolo[1,5-*a*]pyrimidine turned out to be the most promising core for future SAR studies.

## 2. Results and Discussion

### 2.1. Chemistry

The final compounds of our design were obtained in three different multistage approaches. The appropriate aminopyrazole derivatives (available commercially or synthesized) were used as the respective starting materials to provide the final inhibitors utilizing mainly the Buchwald–Hartwig reaction, the Suzuki coupling, or the Dess–Martin periodinane oxidation as the crucial synthetic steps.

#### 2.1.1. Synthesis of Compounds **5–3**

2-Methyl pyrazolo[1,5-*a*]pyrimidine derivatives were obtained in a multi-step reaction according to Figure 1. 5-Amino-3-methylpyrazole was reacted with diethyl malonate in the presence of a base (sodium ethanolate) to obtain dihydroxy-heterocycle **1** (89% yield). Then, 2-methylpyrazolo[1,5-a]pyrimidine-5,7-diol (**1**) was subjected to the chlorination reaction with phosphorus oxychloride to give 5,7-dichloro-2-methylpyrazolo[1,5-*a*]pyrimidine (**2**) (61% yield). Structure **3** was prepared from **2** in a nucleophilic substitution reaction using morpholine in the presence of potassium carbonate at room temperature (94% yield). The selectivity of the reaction results from the strong reactivity of the chlorine atom at position 7 of the pyrazolo[1,5-*a*]pyrimidine core [50]. 4-{5-Chloro-2-methylpyrazolo[1,5-*a*]pyrimidin-7-yl}morpholine (**3**) is the key intermediate in the preparation of a series of Appendix A. Depending on the R^1^ substituent, the final compounds were prepared from **3** using two types of coupling reactions: either the Buchwald–Hartwig or the Suzuki coupling reaction. Benzimidazole derivatives **5**–**7** were synthesized by carrying out the three-step reaction: again, the Buchwald–Hartwig reaction (average yield of 61%), amidation, following the final cyclization step. The corresponding amides **5**–**7** were prepared in the presence of EDCI and HOBt from the appropriate carboxylic acids and amine **4,** resulting from the Buchwald–Hartwig synthesis by the heterocycle ring closure in the presence of glacial acetic acid. Since this synthetic route requires no intermediate purification, the observed yields are satisfactory in the 74–77% range. A separate synthetic route was chosen for compound **9**, obtained in two steps by the Buchwald–Hartwig reaction with a masked aminopyrazole (54% yield), followed by the final deprotection of intermediate **8** (89% yield). Derivatives **10**–**13** were prepared by the Suzuki reaction of compound **3** with the respective esters or boronic acids in the presence of a palladium catalyst with yields in the range of 55–61%.

#### 2.1.2. Synthesis of Compounds **23**–**45**

The synthesis of Appendix A was more complicated and required several additional steps. The first three steps leading to compound **16** were performed based on the available literature data [51,52,53,54]. Initially, the reaction of benzyl alcohol with ethyl bromoacetate in the presence of sodium hydride gave the corresponding ether **14** (Figure 2) with a 76% yield. Then the beta-ketoester derivative **15** was prepared by reaction with acetonitrile under basic conditions using 2,5 M n-butyllithium solution at a lower temperature of −78 °C. Compound **15** was subsequently condensed with hydrazine to give the corresponding aminopyrazole derivative **16** in satisfying 87% yield after two steps, as depicted in Figure 2. The experiences gained in the previous synthetic route could be successfully extrapolated to accomplish the next four steps of the synthesis. Reaction of diethyl malonate with the aminopyrazole derivative **16** gave 2-[(benzyloxy)methyl]pyrazolo[1,5-*a*]pyrimidine-5,7-diol (**17**, 84% yield). Chlorination of **17** with phosphorus oxychloride provided the corresponding dichloro-derivative: 2-[(benzyloxy)methyl]-5,7-dichloropyrazolo[1,5-*a*]pyrimidine (**18**) in 38% yield. A selective and efficient (92% yield) substitution of the C(7)-chlorine atom in the heteroaromatic core with morpholine gave the analog of **3** (Figure 1) as intermediate **19**. Applying the Suzuki coupling conditions to **19** with indole-4-boronic acid pinacol ester led to benzyl masked alcohol **20** in 83% yield. Classical deprotection conditions (gaseous hydrogen over palladium catalyst on activated charcoal) of the benzyloxy group provided compound **21** in 66% yield. The subsequent oxidation reaction of primary alcohol **21** to the crucial aldehyde **22** was easily accomplished using the Dess–Martin reagent (Figure 2) with a yield of 78%. A series of the reductive amination reactions utilizing compound **22** as a key intermediate with the appropriate cyclic amines gave additional contributors (**23** to **45**) to the growing library of PI3K*δ* inhibitors in un-optimized yields varying from 25 to 93%.

#### 2.1.3. Synthesis of Compounds **49–51** and **53–55**

An essential intermediate **19** (Figure 2) was also successfully used to prepare another set of compounds functionalized at the C(5) position to explore more deeply the structure-activity relationship of this particular core. The synthesis of another subset of substituted pyrazolo[1,5-*a*]pyrimidines is shown in Figure 3. Due to the same reaction types, the synthesis pathways of examples Appendix A were similar to the synthesis of the previous compounds (**23****–45**, Figure 2), the difference being the order of the Suzuki reaction and the reductive amination reaction sequence in the multistage synthesis pathway. After deprotection of the hydroxyl group of **19**, compound **46** was oxidized to aldehyde **47** (Figure 3). The following steps included a reductive amination reaction with the carefully selected, based on in silico calculations, amines: (2-(4-piperidyl)-2-propanol or *N*-*t*-butylpiperazine followed by a Suzuki coupling to provide Appendix A, respectively (Figure 3).

### 2.2. Docking Study

Several approaches have been described leading to various structural docking theories explaining the selectivity of PI3K*δ* inhibitors [25,27]. Opening the specificity pocket between the two amino acids, Trp-812 and Met-804, and adopting the appropriate shape within the protein combined with additional correlations, allows the identification of much more selective PI3K*δ* inhibitors from all PI3K Class I isoforms [25,27,34]. It was reported that there are many meaningful interactions between ligand and protein in the enzyme’s active site [6,24,27]. First is the hydrogen bond of the morpholine from pyrazolo[1,5-*a*]pyrimidine derivative in the hinge-binding motif [6,24,25,26]. More precisely, the hydrogen bonding between the oxygen atom from the morpholine mentioned above the ring and amino acid Val-828 was crucial in the hinge region. It has been suggested that indole derivatives in the C(5) position of the core of pyrazolo[1,5-*a*]pyrimidine may form an additional hydrogen bond with Asp-787 (another important interaction in many selective inhibitors, most with the affinity pocket) [25]. For this reason, indole heterocycle-based inhibitors are more selective for PI3K*δ* than other PI3K isoforms. In addition, a suitable substituent of this structure, which can extend into the solvent, can improve the solubility, ADME properties, and potency of the final compounds [25].

Our work is focused on the pyrazolo[1,5-*a*]pyrimidine scaffold and appropriate further optimization with different C(5) substituents.

An example of our approach showing the possible binding site of compound **13** with the kinase is presented in Figure 2. The docking procedure utilizes the PI3K*δ* protein (PDB: 2WXP) and the Auto-Dock Vina program [55]. Compound **13** (magenta) binds similarity to protein as referent compound GDC-0941 (orange, Figure 2). More specifically, the oxygen atom in the morpholine ring forms a hydrogen bond with the amino acid (Val-828) in the hinge region of the enzyme (the importance of this interaction has been explained before). Moreover, the indole system’s hydrogen atom (NH) is involved in forming the hydrogen bond with the carbonyl oxygen in Asp-787 in the affinity pocket of the kinase (Figure 2).

Among the structures **24**, **36**, and **37** additional features were found in our in silico model compared to **13** and similars. Compared to compound **23**, higher activity and selectivity can be explained by interactions with the tryptophan shelf (2WXP: Trp-760) in PI3K*δ*, as described by Sutherlin et al. [25]. For those compounds, the distance between the R^2^ substituent and the tryptophan’s indole ring is significantly shorter (Figure 3A). Moreover, the additional hydrogen bond of the hydroxyl group in (2-(piperidin-4-yl) propan-2-ol) (**36**) with Lys-708 was observed (Figure 3B). On the other hand, for a derivative containing *tert*-butylpiperazine (**37**), strong hydrophobic interactions with tryptophan (Trp-760) were found, which may cause the withdrawal of the indole ring of **37** from the enzyme affinity pocket. Most likely, this situation is observed due to the lack of interaction with tyrosine (Tyr-813) and aspartic acid (Asp-787) in the mentioned pocket (Figure 3B).

### 2.3. Biological Evaluation

#### In Vitro PI3 Kinase Inhibition Assays

To verify whether the 7-(morpholin-4-yl) pyrazolo[1,5-*a*] pyrimidine system can inhibit PI3*δ* kinase, the synthesized compounds **6****–13** were tested for inhibition of selected PI3K*δ* and PI3K*α* kinases activity. Enzymatic tests have been used, and the results are presented in Table 1.

The activity of these compounds ranged from 45 µM to 0.5 µM for the PI3K *δ* isoform and from over 60 µM to 1.06 µM for the PI3K*α* isoform, and thus the *α*/*δ* selectivity ranged from 1 to 30 (Table 1). Among all benzimidazole derivatives synthesized, the most promising activity with the low PI3K*δ* IC_50_ value was measured for compound **7** (IC_50_ = 0.47 µM) (Table 1). On the other hand, compounds **5** and **6**, keeping benzimidazole derivatives within their structures, show significantly lower activity against the PI3K*δ* isoform than compound **7** (IC_50_ value of 3.56 µM and 2.30 µM, respectively), regardless of better selectivity against the PI3K*α* isoform (*α*/*δ*) (9.9 for **5** and 11 for **6**). We observed that compounds with a monocyclic 5-or 6-membered heteroaromatic ring (**9**–**11**) turned out to be less active and thus showed a lower enzyme inhibition potential than the other bicyclic structures. Structures **12** and **13** bearing conjugated bicyclic system as the R^1^ substituent presented a similar activity to the benzimidazole derivatives. The most active were compounds having R^1^ substituents in the form of 2-difluoromethylbenzimidazole (**7**) and indole (**13**). Specifically, their IC_50_ value against PI3K*δ* was 0.475 µM and 0.772 µM, respectively. Due to the much better *α*/*δ* selectivity of compound **13** over compound **7** (*α*/*δ* = 30 and *α*/*δ* = 2.2, respectively), we have chosen the indole derivatives for further optimization.

Compared to compound **13**, significantly more sterically demanding derivatives were designed and synthesized as the next optimization step. While the indole fragments were preserved, many different cyclic amines were linked to the scaffold core through a methylene linkage as an R^2^ substituent (Table 2).

The synthesis of the new group of pyrazolo[1,5-*a*] pyrimidine derivatives (depicted in Figure 2) required additional steps related to the functionalization of the C(2)-position of the heteroaromatic core. Firstly, a group of derivatives with differing sizes of heterocycle rings and different chemical properties of substituents (**23**–**31**) was synthesized (Table 2). We noted that structures containing monocyclic five-membered rings (**25**–**26**) and morpholine (**28**) turned out to be less potent PI3K*δ* inhibitors than compound **13** (Table 1). The mesylpiperazine group present in the GDC0941 Reference [34] did not significantly improve the activity of structurally similar compound **23** from our library (the IC_50_ value of that example for PI3K*δ* and PI3K*α* was 0.4 µM and 2.35 µM, respectively). Urea-derivatives, **30** and **31**, also showed moderate activity. The most potent compounds in this group (Table 2) turn out to be the analogs of *N*,*N*-dimethyl-4-aminopiperidine (**24**), and 4-(*N*-methylpiperazin-1ylo)piperidine (**29**). Both, **24** and **29,** showed promising inhibitory activity against PI3K*δ* (37 nM and 52 nM respectively) and selectivity against other isoforms (*α*/*δ* = 172; *β*/*δ* = 389; *γ*/*δ* = 1332 for **24** and *α*/*δ* = 301 for **29**). Careful structural analysis around the R^2^ substituent of the examples provided in Table 2 led us to several conclusions. Relatively modest activities of the compounds containing the methyl group, aromatic ring, or ester group at the C(4)-position of the heterocyclic ring misled us towards the synthesis of piperazine and piperidine analogs(**32**–**45**) (Table 2). Moreover, the presence of the second ring within the R^2^ substituent (compounds **39**–**40** and **42**–**45**) did not improve PI3K*δ* activity compared to previously obtained compounds **24** or **29**. Finally, only large aliphatic substituents within piperazine or piperidine rings gain the PI3K*δ* potency and respective selectivity.

We observed that the best results were achieved for two compounds being the representatives of two different modifications. More specifically 2-(piperidin-4-yl) propan-2-ol (compound **36** of piperidine modification series) and *N*-*tert*-butylpiperazine (compound **37** of piperazine modification series) exhibit high activities towards the PI3K*δ* (IC_50_ = 6.6 and 13.0 nM, respectively) and appreciable selectivities towards other isoforms (*α*/*δ* = 1217; *β*/*δ* = 332; *γ*/*δ* = 1223 for **36** and *α*/*δ* = 1889; *β*/*δ* = 829; *γ*/*δ* > 9091 for **37**; Table 2). As the hit to lead optimization route continued, several indole and azaindole derivatives at the C(5) position were introduced to the existing scaffold. While preserving the most active amino groups, we prepared the piperidine derivatives series (summarized in Table 3) and piperazine derivatives series (covered in Table 4). From all the synthesized structures, the *N*-*tert*-butylpiperazine derivatives (**37**, **53**, **54**, **55**, Table 4) show the highest PI3K*δ* activity, greater than the piperidyl–propanol analogs shown in Table 3 (**36**, **49**, **50**, **51**). The presence of the fluorine atom in the C(5)-position of the indol fragment causes a slight decrease in activity against the PI3K*δ* isoform in both groups without affecting the selectivity toward other isoforms. The introduction of the nitrogen atom to the indole ring at position 7 caused a slight decrease in the activity of compound **51** (Table 3), which was almost doubled in the case of **55** (Table 4). Moreover, slight decreases in activity related to the PI3K*α* isoform were observed for these structures. An introduction of a nitrogen atom in the 6-position of the indole caused a decrease in activity derivate **50** but a 10-fold improvement for **54**. Decreased selectivity against the PI3K*α* isoform was also observed for the azaindole structures (**50**, **51**, **53**, **54**) despite the good activity in the nanomolar range (IC_50_ value: 2.8–45 nM).

We have found that two compounds: **37** and **54**, from the entire synthesized library showed the best activity and selectivity for PI3K*δ.* Based on all parameters, these structures showed the highest selectivity, the lowest IC_50_ values, and the most promising other parameters [15]. Consequently, those two selected examples were tested by flow cytometry towards the proliferation of B lymphocytes capabilities. Both showed very high potency in inhibiting B cell proliferation with IC_50_ values of 20 nM and 19 nM, respectively (Table 5). Moreover, compound **54** had better kinetic solubility at pH 7.4 than compound **37** (>500 and 444 µM respectively) (Table 5). We also observed that the presence of nitrogen atom in the 6-azaindole ring of **54** molecule results in higher metabolic stability in murine and human microsomes (for details, see Table 5).

## 3. Materials and Methods

### 3.1. Chemistry

#### 3.1.1. General Information

Chemicals (at least 95% purity) were purchased from ABCR (Karlsruhe, Germany), Acros (Geel, Belgium), Alfa Aesar (Haverhill, MA, USA), Combi-Blocks (San Diego, CA, USA), Fluorochem (Hadfield, UK), Fluka (Charlotte, NC, USA), Merck (Rahway, NJ, USA), and Sigma Aldrich (St. Louis, MO, USA) and were used without additional purification. Solvents were purified according to standard procedures if required. Air or moisture-sensitive reactions were carried out under an argon atmosphere. All reaction progresses were routinely checked by thin-layer chromatography (TLC). TLC was performed using silica gel coated plates (Kieselgel F254) and visualized using UV light. Flash chromatography was performed using Merck silica gel 60 (230–400 mesh ASTM). ^1^H NMR spectra were acquired on a Varian Inova 300 MHz NMR spectrometer, JOEL JNMR-ECZS 400 MHz spectrometer, JOEL JNMR-ECZR 600 MHz spectrometer, and Bruker DRX 500 NMR spectrometer with ^1^H being observed at 300 MHz, 400 MHz, 600 MHz, and 500 MHz, respectively. ^13^C NMR spectra were recorded similarly at 75 MHz, 101 MHz, 151 MHz, and 126 MHz, frequencies for ^13^C, respectively. Due to the poor solubility of some final compounds, usual characterization by ^13^C NMR was omitted. Chemical shifts for ^1^H and ^13^C NMR spectra were reported in *δ* (ppm) using tetramethylsilane as an internal standard or according to the residual undeuterated solvent signal (2.50 ppm for DMSO-*d*_6_, and 7.26 ppm for CDCl_3_). The abbreviations for spin interaction coupled ^1^H signals are as follows: s (singlet), d (doublet), t (triplet), m (multiplet), dd (doublet of doublets), dt (doublet of triplet), q (quartet). Coupling constants (J) are expressed in Hertz. Mass spectra (Atmospheric Pressure Ionization Electrospray, API-ES, and Electrospray Ionization, ESI-MS) were obtained using Agilent 6130 LC/MSD spectrometer or Agilent 1290 UHPLC coupled with Agilent QTOF 6545 mass spectrometer.

#### 3.1.2. Synthesis

Procedure for 5,7-dihydroxy-2-methylpyrazolo[1,5-*a*]pyrimidine (**1**)

To the flask with sodium ethoxide solution (obtained from sodium (4.73 g, 0.21 mol) and ethanol (175 mL) a solution of 3-amino-5-methylpyrazole (10.0 g, 0.10 mol) in ethanol (100 mL) and diethyl malonate (23.5 mL, 0.15 mol) were added. The reaction was carried out at reflux for 24 h. The reaction mixture was cooled to room temperature, and then the solvent was evaporated under reduced pressure. The residue was dissolved in 1200 mL of water and acidified with concentrated hydrochloric acid to a pH of about 2. Creamy solid precipitated from the solution was filtered off, washed, and dried. The title compound **1** (15.2 g, 0.08 mol) was obtained as an off-white solid with 89% yield. MS-ESI: *m*/*z* calcd for C_7_H_7_N_3_O_2_ [M+Na]^+^: 188.04; found 187.9.

Procedure for 5,7-dichloro-2-methylpyrazolo[1,5-*a*]pyrimidine (**2**)

To the cooled to 0 °C POCl_3_ (90 mL, 0.963 mol), compound **1** (15.2 g, 0.092 mol) was added. The reaction was carried out at reflux for 24 h. The reaction mixture was cooled to room temperature and poured into the water with ice. The mixture was quenched with a 6 M sodium hydroxide solution to pH 6. The aqueous phase was extracted with ethyl acetate, and after separation, the organic phase was dried with anhydrous sodium sulfate. After filtration of the drying agent and evaporation of the solvent, the residue was purified by column chromatography (0–40% ethyl acetate gradient in heptane) to give compound **2** (11.4 g, 0.056 mol) obtained as an off-white solid with 61% yield. ^1^H NMR (300 MHz, CDCl_3_) *δ*: 6.90 (s, 1H, Ar-H), 6.53 (s, 1H, Ar-H), 2.56 (s, 3H, CH_3_). MS-ESI: *m*/*z* calcd for C_7_H_5_Cl_2_N_3_ [M+H]^+^: 201.99; found 201.9.

Procedure for 5-chloro-2-methyl-7-morpholin-4-yl-pyrazolo[1,5-*a*]pyrimidine (**3**)

To the solution of compound **2** (2.0 g, 9.9 mmol) in acetone (50 mL), potassium carbonate (1.64 g, 11.9 mmol), and morpholine (1.35 mL, 15.5 mmol) were added. The reaction was carried out at room temperature for 1.5 h. Then water (100 mL) was added to the reaction mixture, and the precipitated white solid was filtered off. The obtained solid was washed with water (50 mL) and water/acetone mixture (2/1, *v*/*v*) (50 mL), then dried. Compound **3** (2.36 g, 0.09 mol) was obtained as a white solid with 94% yield. ^1^H NMR (300 MHz, CDCl_3_) *δ*: 6.29 (s, 1H, Ar-H), 6.01 (s, 1H, Ar-H), 4.00–3.92 (m, 4H, morph.), 3.81–3.72 (m, 4H, morph.), 2.46 (s, 3H, CH_3_). MS-ESI: *m*/*z* calcd for C_11_H_13_ClN_4_O [M+H]^+^: 253.09; found 253.0.

Procedure for *N*-(2-methyl-7-(morpholin-4-yl)pyrazolo[1,5-*a*]pyrimidin-5-yl)benzene-1,2-diamine (**4**)

The mixture of compound **3** (1.0 g, 3.96 mmol), benzene-1,2-diamine (1.31 g, 11.9 mmol), cesium carbonate (3.87 g, 11.9 mmol), tris(dibenzylideneacetone)dipalladium(0) (0.181 g, 0.20 mmol), 9,9-dimethyl-4,5-bis(diphenylphosphine)xanthene (0.229 g, 0.40 mmol) and dry toluene (40 mL) were introduced to the reaction Schlenk flask. The mixture was flushed with argon and stirred at 110 °C for 24 h. After cooling to room temperature, the reaction mixture was filtered through Celite^®^, and the solid was washed with ethyl acetate. The filtrate was concentrated under reduced pressure using an evaporator. The residue was resolved and purified by column chromatography (50–100% ethyl acetate gradient in heptane) to give the title compound **4** (0.78 g, 2.4 mmol) with 61% yield. ^1^H NMR (300 MHz, CDCl_3_) *δ* 7.23–7.17 (m, 1H, Ar-H), 7.16–7.09 (m, 1H, Ar-H), 6.88–6.76 (m, 2H, Ar-H), 6.37 (s, 1H, Ar-H), 5.92–5.86 (m, 1H), 5.30 (s, 1H), 4.01–3.81 (m, 4H, morph.), 3.58–3.45 (m, 4H, morph.), 2.39 (s, 3H, CH_3_). MS-ESI: *m*/*z* calcd for C_17_H_20_N_6_O [M+H]^+^: 325.18; found 325.1.

General Procedure for the Synthesis of Benzimidazole Derivatives (**5**–**7**)

In the solution of compound **4** (1.0 eq) dissolved in dry DCM (10 mL/1g of compound **4**), the carboxylic acid (2.0 eq), HOBt × H_2_O (1.2 eq), EDCI × HCl (2.4 eq), and TEA (3.0 eq) were added. The whole reaction mixture was stirred at room temperature for 48 h. To the reaction, mixture water was added, and organic and water phases were separated. The aqueous phase was washed three times with DCM. Combined organic phases were dried over anhydrous sodium sulfate. After the drying agent was filtered off and the solvent evaporated, the reaction mixture was dissolved in glacial acetic acid. The reaction mixture was refluxed for 24 h. Then the reaction mixture was cooled and concentrated under reduced pressure. The residue was diluted with water and neutralized with a saturated sodium bicarbonate solution. The aqueous phase was extracted three times with ethyl acetate. Combined organic phases were dried over sodium sulfate. Once the drying agent was filtered off, the solvent was evaporated under reduced pressure using an evaporator. The reaction mixture was purified by column chromatography.

2-methyl-5-(2-methylbenzimidazol-1-yl)-7-(morpholin-4-yl)pyrazolo[1,5-*a*]pyrimidine (**5**)

Compound **5** was prepared from compound **4** (0.20 g, 0.62 mmol), acetic acid (70 μL, 74 mg, 1.23 mmol), HOBt (0.10 g, 0.74 mmol), EDCI (0.28 g, 1.48 mmol), TEA (0.26 mL, 0.19 g, 1.85 mmol) and DCM (6.0 mL). The crude product was purified by flash chromatography to give **5** (0.16 g, 0.46 mmol) as a light yellow solid with 73% yield. ^1^H NMR (300 MHz, CDCl_3_) *δ* 7.78–7.72 (m, 1H, Ar-H), 7.50–7.45 (m, 1H, Ar-H), 7.34–7.22 (m, 2H, Ar-H), 6.42 (s, 1H, Ar-H), 6.16 (s, 1H, Ar-H), 4.03–3.97 (m, 4H, morph.), 3.88–3.82 (m, 4H, morph.), 2.76 (s, 3H, CH_3_), 2.53 (s, 3H, CH_3_). ^13^C NMR (75 MHz, CDCl_3_) *δ* 155.0, 151.6, 151.2, 150.4, 148.5, 142.7, 134.5, 123.0, 122.9, 119.4, 110.4, 96.1, 87.5, 66.2, 48.4, 15.6, 14.8. MS-ESI: *m*/*z* calcd for C_19_H_20_N_6_O [M+H]^+^: 349.18; found 349.1.

2-methyl-5-(2-ethylbenzimidazol-1-yl)-7-(morpholin-4-yl)pyrazolo[1,5-*a*]pyrimidine (**6**)

Compound **6** was prepared from compound **4** (0.20 g, 0.62 mmol), propionic acid (92 μL, 91 mg, 1.23 mmol), HOBt (0.10 g, 0.74 mmol), EDCI (0.28 g, 1.48 mmol), TEA (0.26 mL, 0.19 g, 1.85 mmol) and DCM (6.0 mL). The crude product was purified by flash chromatography to give **6** (0.17 g, 0.47 mmol) as a white solid with 75% yield. ^1^H NMR (300 MHz, CDCl_3_) *δ* 7.83–7.76 (m, 1H, Ar-H), 7.47–7.41 (m, 1H, Ar-H), 7.34–7.21 (m, 2H, Ar-H), 6.42 (s, 1H, Ar-H), 6.15 (s, 1H, Ar-H), 4.03–3.97 (m, 4H, morph.), 3.88–3.82 (m, 4H, morph.), 3.11 (q, *J =* 7.5 Hz, 2H, CH_2_), 2.53 (s, 3H, CH_3_), 1.41 (t, *J =* 7.5 Hz, 3H, CH_3_). ^13^C NMR (75 MHz, CDCl_3_) *δ* 156.2, 155.0, 151.2, 150.4, 148.4, 142.7, 134.6, 123.0, 122.7, 119.5, 110.2, 96.2, 87.7, 66.2, 48.4, 22.1, 14.8, 11.9. MS-ESI: *m*/*z* calcd for C_20_H_22_N_6_O [M+H]^+^: 363.19; found 363.1.

2-methyl-5-(2-difluoromethylbenzimidazol-1-yl)-7-(morpholin-4-yl)pyrazolo[1,5-*a*]pyrimidine (**7**)

Compound **7** was prepared from compound **4** (0.20 g, 0.62 mmol), difluoroacetic acid (77 μL, 0.12 g, 1.23 mmol), HOBt (0.10 g, 0.74 mmol), EDCI (0.28 g, 1.48 mmol), TEA (0.26 mL, 0.19 g, 1.85 mmol) and DCM (6.0 mL). The crude product was purified by flash chromatography to give **7** (0.18 g, 0.47 mmol) as a white solid with 76% yield. ^1^H NMR (300 MHz, CDCl_3_) *δ* 7.92 (d, *J =* 7.1 Hz, 1H, Ar-H), 7.65 (d, *J =* 7.4 Hz, 1H, Ar-H), 7.47–7.38 (m, 2H, Ar-H), 7.24 (t, *J =* 26.8 Hz, 1H, CHF_2_), 6.42 (s, 1H, Ar-H), 6.28 (s, 1H, Ar-H), 4.02–3.97 (m, 4H, morph.), 3.92–3.87 (m, 4H, morph.), 2.53 (s, 3H, CH_3_). MS-ESI: *m*/*z* calcd for C_19_H_18_F_2_N_6_O [M+H]^+^: 385.16; found 385.0.

Procedure for 1-*tert*-butyl-3-methyl-*N*-[2-methyl-7-(morpholin-4-yl)pyrazolo[1,5-*a*]pyrimidin-5-yl]-1*H*-pyrazol-5-amine (**8**)

The mixture of compound **3** (0.64 g, 2.53 mmol), 1-*tert*-butyl-3-methyl-1*H*-pyrazol-5-amine (0.59 g, 3.86 mmol), cesium carbonate (1.70 g, 5.16 mmol), tris(dibenzylideneacetone)dipalladium(0) (0.13 g, 0.12 mmol), 9,9-dimethyl-4,5-bis(diphenylphosphine)xanthene (0.15 g, 0.25 mmol) and dry toluene (30 mL) were introduced to the reaction Schlenk flask. The whole mixture was flushed with argon and stirred at 100 °C for 18 h. After cooling to room temperature, the reaction mixture was filtered through the Celite^®^, and the solid was washed with CHCl_3_ (50 mL). The filtrate was concentrated under reduced pressure. The residue was resolved on a chromatographic column (amine-functionalized silica gel) (0–10% ethyl acetate gradient in heptane) to give compound **8** (0.51g, 1.38 mmol) with 54% yield. ^1^H NMR (300 MHz, CDCl_3_) *δ* 7.02 (s, 1H, Ar-H), 6.01 (s, 1H, Ar-H), 5.77 (s, 1H), 3.96–3.86 (m, 4H, morph.), 3.59–3.49 (m, 4H, morph.), 2.38 (s, 3H, CH_3_), 2.29 (s, 3H, CH_3_), 1.60 (s, 9H, *t*-Bu.). ^13^C NMR (75 MHz, CDCl_3_) *δ* 156.9, 154.3, 151.9, 151.0, 146.1, 137.3, 104.6, 92.1, 79.2, 66.5, 59.8, 48.8, 30.3, 15.0, 14.6. MS-ESI: *m*/*z* calcd for C_19_H_27_N_7_O [M+H]^+^: 370.24; found 370.1.

Procedure for 3-methyl-*N*-[2-methyl-7-(morpholin-4-yl)pyrazolo[1,5-*a*]pyrimidin-5-yl]-1*H*-pyrazol-5-amine (**9**)

Compound **8** (0.20 g, 0.545 mmol), trifluoroacetic acid (1.0 mL), and water (4.0 mL) were refluxed for 20 h. Then, the reaction mixture was cooled to room temperature, water (10 mL) was added and the whole mixture was alkalized with saturated sodium carbonate solution (12 mL). Precipitation was observed and obtained solid was filtered off, washed with water (5 mL), and dried. The title compound **9** (0.15 g, 0.48 mmol) was isolated as a white solid with 89% yield. ^1^H NMR (400 MHz, DMSO-*d*_6_) *δ* 11.86 (s, 1H, NH), 9.41 (s, 1H, NH), 6.30 (s, 1H, Ar-H), 6.06 (s, 1H, Ar-H), 5.85 (s, 1H, Ar-H), 3.80–3.78 (m, 4H, morph.), 3.52–3.50 (m, 4H, morph.), 2.27 (s, 3H, CH_3_), 2.20 (s, 3H, CH_3_). ^13^C NMR (101 MHz, DMSO-*d*_6_) *δ* 153.6, 151.6, 150.7, 150.1, 95.1, 91.4, 81.8, 65.6, 48.0, 14.4, 10.9. MS-ESI: *m*/*z* calcd for C_15_H_19_N_7_O [M+H]^+^: 314.17; found 314.1.

General Procedure for the Suzuki Reaction

To the solution of compound **3** (1.0 eq) dissolved in 1,2-dimethoxyethane (DME) (10 mL/1 g of compound **3**), boronic acid pinacol ester or boronic acid (1.5 eq), tetrakis(triphenylphosphino)palladium (0) (0.2 eq) and 2M aqueous sodium carbonate solution (2.0 eq) were added. The reaction mixture was refluxed overnight. Then, the reaction mixture was cooled to room temperature, filtered through the pad of Celite^®^, and obtained solid washed with ethyl acetate. The filtrate was concentrated under reduced pressure using an evaporator and the residue was purified by column chromatography.

2-methyl-7-(morpholin-4-yl)-5-(1*H*-pyrazol-4-yl)pyrazolo[1,5-*a*]pyrimidine (**10**)

Synthesized from compound **3** (0.15 g, 0.594 mmol), 4-(4,4,5,5-tetramethyl-1,3,2-dioxaborolan-2-yl)-1*H*-pyrazole-1-carboxylic acid tert-butyl ester (0.26 g, 0.890 mmol), tetrakis(triphenylphosphine)palladium(0) (0.14 g, 0.119 mmol), 2M aqueous sodium carbonate solution (0.59 mL, 1.19 mmol) and DME (6 mL). The crude product was purified by flash chromatography (0–100% ethyl acetate gradient in heptane) to give **10** (0.095 g, 0.33 mmol) with 56% yield. ^1^H NMR (300 MHz, DMSO-*d*_6_) *δ* 13.18 (s, 1H, NH), 8.49 (s, 1H, Ar-H), 8.13 (s, 1H, Ar-H), 6.60 (s, 1H, Ar-H), 6.24 (s, 1H, Ar-H), 3.90–3.78 (m, 4H, morph.), 3.78–3.63 (m, 4H, morph.), 2.37 (s, 3H, CH_3_). ^13^C NMR (75 MHz, DMSO-*d*_6_) *δ* 152.8, 151.7, 151.4, 149.8, 138.1, 128.7, 121.7, 93.9, 89.3, 65.9, 48.3, 14.7. MS-ESI: *m*/*z* calcd for C_14_H_16_N_6_O [M+H]^+^: 285.15; found 284.9.

2-methyl-7-(morpholin-4-yl)-5-(2-aminopyridin-5-yl)pyrazolo[1,5-*a*]pyrimidine (**11**)

Synthesized from compound **3** (0.10 g, 0.396 mmol), 2-aminopyridine-5-boronic acid pinacol ester (0.14 g, 0.594 mmol), tetrakis(triphenylphosphine)palladium(0) (91 mg, 0.079 mmol), 2M aqueous sodium carbonate solution (0.40 mL, 0.791 mmol) and DME (4 mL). the crude product was purified by flash chromatography (0–100% ethyl acetate gradient in heptane) to give **11** (0.075 g, 0.032 mol) with 61% yield. ^1^H NMR (300 MHz, CDCl_3_+CD_3_OD) *δ* 8.58 (d, *J =* 1.8 Hz, 1H), 8.11 (dd, *J =* 8.8, 1.8 Hz, 1H), 6.68 (d, *J =* 8.8 Hz, 1H, Ar-H), 6.47 (s, 1H, Ar-H), 6.34 (s, 1H), 4.04–3.98 (m, 4H, morph.), 3.80–3.75 (m, 4H, morph.), 2.49 (s, 3H, CH_3_). ^13^C NMR (75 MHz, CDCl_3_+CD_3_OD) *δ* 146.6, 136.8, 108.8, 100.2, 94.7, 88.7, 74.8, 70.2, 66.1, 29.5, 16.55, 14.0. MS-ESI: *m*/*z* calcd for C_16_H_18_N_6_O [M+H]^+^: 311.16; found 311.0.

2-methyl-7-(morpholin-4-yl)-5-(1*H*-indazole-4-yl)pyrazolo[1,5-*a*]pyrimidine (**12**)

Synthesized from compound **3** (0.10 g, 0.396 mmol), 1*H*-indazole-4-boronic acid (0.10 g, 0.594 mmol), tetrakis(triphenylphosphine)palladium(0) (91 mg, 0.079 mmol), 2M aqueous sodium carbonate solution (0.40 mL, 0.791 mmol) and DME (4 mL). The crude product was purified by flash chromatography (0–100% ethyl acetate gradient in heptane) to give **12** (0.077 g, 0.23 mmol) with 58% yield. ^1^H NMR (300 MHz, CDCl_3_) *δ* 8.76–8.74 (m, 1H, NH), 7.69–7.65 (m, 1H), 7.61–7.57 (m, 1H, Ar-H), 7.53–7.46 (m, 1H), 6.59 (s, 1H, Ar-H), 6.50 (s, 1H, Ar-H), 4.06–3.98 (m, 4H, morph.), 3.85–3.76 (m, 4H, morph.), 2.54 (s, 3H, CH_3_). ^13^C NMR (75 MHz, CDCl_3_) *δ* 156.8, 154.8, 151.9, 150.6, 141.2, 135.8, 132.3, 128.9, 126.9, 121.0, 111.7, 96.3, 91.2, 66.6, 48.7, 15.2. MS-ESI: *m*/*z* calcd for C_18_H_18_N_6_O [M+H]^+^: 335.16; found 335.1.

2-methyl-7-(morpholin-4-yl)-5-(1*H*-indole-4-yl)pyrazolo[1,5-*a*]pyrimidine (**13**)

Synthesized from compound **3** (0.10 g, 0.404 mmol), indole-4-boronic acid pinacol ester (0.15 g, 0.606 mmol), tetrakis(triphenylphosphine)palladium(0) (93 mg, 0.081 mmol), 2M aqueous sodium carbonate solution (0.40 mL, 0.80 mmol) and DME (5 mL). The crude product was purified by flash chromatography (0–50% ethyl acetate gradient in heptane) to give **13** (0.074 g, 0.22 mmol) with 55% yield. ^1^H NMR (300 MHz, CDCl_3_) *δ* 8.68 (s, 1H, NH), 7.63–7.55 (m, 1H, Ar-H), 7.48–7.39 (m, 1H, Ar-H), 7.33–7.21 (m, 2H, Ar-H), 7.10–7.04 (m, 1H, Ar-H), 6.61 (s, 1H, Ar-H), 6.45 (s, 1H, Ar-H), 4.05–3.93 (m, 4H, morph.), 3.82–3.70 (m, 4H, morph.), 2.52 (s, 3H, CH_3_). ^13^C NMR (75 MHz, CDCl_3_) *δ* 158.4, 154.1, 151.7, 150.1, 136.6, 131.3, 125.9, 125.4, 121.9, 120.2, 112.6, 102.6, 95.5, 92.1, 66.3, 48.4, 14.8. MS-ESI: *m*/*z* calcd for C_19_H_19_N_5_O [M+H]^+^: 334.17; found 334.0.

Procedure for Ethyl 2-benzyloxyacetate (**14**)

To the suspension of 60% NaH (21.8 g, 0.545 mol) in dry toluene (1000 mL), benzyl alcohol (47 mL, 0.454 mol) was added dropwise over 30 min. The whole mixture was stirred at room temperature for 4 h. The suspension was cooled in a water-ice bath and ethyl bromoacetate (66 mL, 0.595 mol) was added dropwise for 45 min. The reaction mixture was heated to room temperature and stirred for one h. The whole mixture was poured onto ice water (1200 mL) acidified with concentrated hydrochloric acid (10 mL) to pH 4. Phases were separated and the aqueous phase was extracted three times with diethyl ether (200 mL). Combined organic phases were washed with brine and dried over anhydrous magnesium sulfate. After filtration of the drying agent, organic solvents were evaporated under reduced pressure. The residue was separated by distillation under reduced pressure to give (66.7 g, 0.34 mol) ethyl 2-benzyloxyacetate (**14**) with 76% yield as a colorless liquid (T_b_ = 104–106°C / 0.7 tor). ^1^H NMR (500 MHz, CDCl_3_) *δ*: 7.39–7.28 (m; 5H, Ar-H), 4.63 (s; 2H, CH_2_), 4.23 (q; *J =* 7.1 Hz; 2H, CH_2_), 4.09 (s; 2H, CH_2_), 1.28 (t; *J =* 7.1 Hz; 3H, CH_3_). MS-ESI: *m*/*z* calcd for C_11_H_14_O_3_ [M+H]^+^: 195.23; found 195.1.

Procedure for 4-benzyloxy-3-oxobutyronitrile (**15**)

A flask filled with dry THF (750 mL) under an argon atmosphere was cooled to −78 °C, then 2.5 M n-BuLi hexane solution (200 mL, 0.5 mol) was added, and after that acetonitrile (28 mL, 0.533 mol) was added dropwise. The whole mixture was stirred at −78 °C for 2 h. The mixture was transferred dropwise to the suspension of ethyl 2-benzyloxyacetate (77.7 g, 0.4 mol) dropwise, and stirring was continued at −78 °C for one h. The reaction was quenched with ammonium chloride solution (500 mL). The reaction mixture was poured onto ice water and acidified with 6 M hydrochloric acid (250 mL) to pH 3. The aqueous phase was extracted twice with diethyl ether (400 mL). Combined organic phases were washed with brine and dried over anhydrous magnesium sulfate. The drying agent was filtered off, and the solvent was evaporated under reduced pressure. Compound **15** was used in the next step without additional purification. MS-ESI: *m*/*z* calcd for C_11_H_11_NO_2_ [M+H]^+^: 190.22; found 190.1.

Procedure for 3-(benzyloxymethyl)-1*H*-pyrazol-5-amine (**16**)

To compound **15** (75.7 g, 0.4 mol, obtained above), ethanol (500 mL) and hydrazine monohydrate (100 mL, 2.1 mol) were added. The mixture was refluxed for 16 h. After concentration, the residue was dissolved with chloroform and dried over anhydrous sodium sulfate. Then, the drying agent was filtered off, and the solvent was evaporated. The crude product was purified by column chromatography (0–5% methanol gradient in ethyl acetate) to give **16** (70.4 g, 0.34 mol) with 87% yield after two steps as a brown oil. ^1^H NMR (300 MHz; CDCl_3_) *δ*: 7.39–7.28 (m; 5H, Ar-H); 5.59 (s; 1H); 4.53 (s; 2H, CH_2_); 4.50 (s; 2H, CH_2_). MS-ESI: *m*/*z* calcd for C_11_H_13_N_3_O [M+H]^+^: 204.25; found 204.1.

Procedure for 2-(benzyloxymethyl)pyrazolo[1,5-*a*]pyrimidin-5,7-diol (**17**)

To the flask containing sodium ethanolate solution (obtained from sodium ethanolate (53 g, 0.74 mol) and ethanol (700 mL)), compound **16** (70.4 g, 0.35 mol) dissolved in ethanol (200 mL) and diethyl malonate (80 mL, 0.53 mol) was added. The reaction was refluxed for 24 h. Then the reaction mixture was cooled to room temperature, and the solvent was evaporated under reduced pressure. The residue was dissolved in water (1200 mL) and acidified with concentrated hydrochloric acid (250 mL). Creamy solid precipitated from the solution was filtered off, washed, and dried to give **17** (79.0 g, 0.27 mol) with 84% yield as a creamy solid. MS-ESI: *m*/*z* calcd for C_14_H_13_N_3_O_3_ [M+Na]^+^: 294.26; found 294.1.

Procedure for 2-(benzyloxymethyl)-5,7-dichloropyrazolo[1,5-*a*]pyrimidine (**18**)

The suspension of compound **17** (30 g, 0.11 mol) in acetonitrile (270 mL) was cooled to 0 °C in a water-ice bath, and POCl_3_ (206 mL, 2.2 mol) was added. The reaction was heated at 80 °C for five h. The reaction mixture was concentrated under reduced pressure to remove acetonitrile and POCl_3_. The residue was poured onto the water with ice and alkalized to pH 5 with saturated sodium hydrogen carbonate solution (350 mL). The aqueous phase was extracted with ethyl acetate, and after separation, the organic phase was dried over anhydrous sodium sulfate. After filtration of the drying agent and evaporation of the solvent, the residue was purified by column chromatography (0–20% ethyl acetate gradient in heptane to give **18** (13 g, 42.3 mmol) with 38% yield as a slightly yellow oil. ^1^H NMR (300 MHz, CDCl_3_) *δ*: 7.41–7.27 (m; 5H, Ar-H); 6.96 (s; 1H, Ar-H); 6.80 (s; 1H, Ar-H); 4.81 (s; 2H, CH_2_); 4.65 (s; 2H, CH_2_). MS-ESI: *m*/*z* calcd for C_14_H_11_Cl_2_N_3_O [M+H]^+^: 309.17; found 308.0.

Procedure for 2-(benzyloxymethyl)-5-chloro-7-(morpholin-4-yl)pyrazolo[1,5-*a*]pyrimidine (**19**)

To the solution of compound **18** (13 g, 42.3 mmol) dissolved in acetone (450 mL), sodium carbonate (5.38 g, 50.8 mmol), and morpholine (6.65 mL, 76.2 mmol) were added. The reaction was carried out at room temperature for 1.5 h. 500 mL of water were added to the reaction mixture, and the precipitated white solid was filtered off. The solid was washed with water (300 mL) and water/acetone mixture (2/1, *v*/*v*) (200 mL), then dried to give **19** (14 g, 39.01 mmol) with 92% yield as a white solid. ^1^H NMR (300 MHz, CDCl_3_) *δ*: 7.41–7.27 (m; 5H, Ar-H); 6.56 (s; 1H, Ar-H); 6.06 (s; 1H, Ar-H); 4.73 (s; 2H, CH_2_); 4.62 (s; 2H, CH_2_); 3.98–3.90 (m; 4H, morph.); 3.82–3.74 (m; 4H, morph.). MS-ESI: *m*/*z* calcd for C_18_H_19_ClN_4_O_2_ [M+H]^+^: 359.83; found 359.2.

Procedure for 2-(benzyloxymethyl)-5-(1H-indol-4-yl)-7-(morpholin-4-yl)pyrazolo[1,5-*a*]pyrimidine (**20**)

To the solution of compound **19** (1.88 g, 5.24 mmol) dissolved in 1,2-dimethoxyethane (DME) (52 mL), indole-4-boronic acid pinacol ester (1.97 g, 7.87 mmol), tetrakis(triphenylphosphino)palladium (0) (0.61 g, 0.52 mmol) and 2M aqueous sodium carbonate solution (5.2 mL) were added. The reaction was refluxed for 16 h. Then, the reaction mixture was cooled to room temperature, filtered through the Celite^®^, and the solid was washed with ethyl acetate (50 mL). The filtrate was concentrated under reduced pressure using an evaporator. The crude product was purified by column chromatography (0–70% ethyl acetate gradient in heptane) to obtain compound **20** (1.91 g, 4.34 mmol) with an 83% yield. ^1^H NMR (300 MHz, CDCl_3_) *δ*: 8.61 (s; 1H); 7.61 (dd; *J =* 7.4; 0.8 Hz; 1H, Ar-H); 7.50–7.23 (m; 8H); 7.13–7.07 (m; 1H, Ar-H); 6.74 (s; 1H, Ar-H); 6.66 (s; 1H, Ar-H); 4.81 (s; 2H, CH_2_); 4.67 (s; 2H, CH_2_); 4.02–3.95 (m; 4H, morph.); 3.81–3.73 (m; 4H, morph.). MS-ESI: *m*/*z* calcd for C_26_H_25_N_5_O_2_ [M+H]^+^: 440.21; found 440.1.

Procedure for [5-(1*H*-indol-4-yl)-7-(morpholin-4-yl)pyrazolo[1,5-*a*]pyrimidin-2-yl]methanol (**21**)

To the solution of compound **20** (5.0 g, 9.1 mmol) in DMF (120 mL) and EtOH (60 mL), 10% Pd/C (11.3 g) and formic acid (100 µL) were added. The reaction was heated to 60 °C under hydrogen pressure for 24 h. After cooling the reaction mixture to room temperature, the catalyst was filtered-off on a Celite^®^, washed with EtOH (50 mL), and the filtrate was then concentrated under reduced pressure using an evaporator. The crude product was purified by column chromatography (0–100% ethyl acetate gradient in heptane) to give **21** (2.08 g, 5.95 mmol) with 66% yield. ^1^H NMR (300 MHz, DMSO-*d*_6_) *δ* 11.36 (s; 1H, NH); 7.70–7.63 (m; 1H, Ar-H); 7.59–7.52 (m; 1H, Ar-H); 7.52–7.46 (m; 1H, Ar-H); 7.28–7.20 (m; 1H, Ar-H); 7.14–7.09 (m; 1H, Ar-H); 6.78 (s; 1H, Ar-H); 6.55 (s; 1H, Ar-H); 5.36 (t; *J =* 6.0 Hz; 1H, OH); 4.66 (d; *J =* 6.0 Hz; 2H, CH_2_); 3.90–3.83 (m; 4H, morph.); 3.83–3.75 (m; 4H, morph.). MS-ESI: *m*/*z* calcd for C_19_H_19_N_5_O_2_ [M+H]^+^: 350.39; found 350.2.

Procedure for 5-(1*H*-indol-4-yl)-7-(morpholin-4-yl)pyrazolo[1,5-*a*]pyrimidin-2-carboxyaldehyde (**22**)

To the solution of compound **21** (0.90 g, 2.58 mmol) in dry DMF(26 mL), Dess–Martin reagent (1.31 g, 3.09 mmol) was added. The whole mixture was stirred at room temperature for one h. The obtained solid was filtered off and then washed with ethyl acetate (35 mL). The obtained solution was concentrated under reduced pressure. The crude product was purified by flash chromatography (0–70% ethyl acetate gradient in heptane) to give **22** (0.70 g, 2.01 mmol) with 78% yield. ^1^H NMR (300 MHz, CDCl_3_) *δ* 10.22 (s; 1H, CHO); 8.47 (s; 1H); 7.66–7.59 (m; 1H, Ar-H); 7.57–7.50 (m; 1H, Ar-H); 7.39–7.29 (m; 2H, Ar-H); 7.18–7.09 (m; 2H, Ar-H); 6.83 (s; 1H, Ar-H); 4.08–4.00 (m; 4H, morph.); 3.86–3.77 (m; 4H, morph.). MS-ESI: *m*/*z* calcd for C_19_H_17_N_5_O_2_ [M+H]^+^: 348.38; found 348.1.

General Procedure for the Reductive Amination Reaction (**23–45**)

To the solution of compound **22** (1.0 eq) in dry DCM (10 mL/1 g of compound **22**), amine derivative (1.2 eq) was added and then stirred at room temperature. After 1 h sodium triacetoxyborohydride (1.5 eq) was added and the mixture was stirred at room temperature for 15 h. To the reaction mixture was added water and phases were separated. The aqueous phase was extracted three times with DCM. Combined organic phases were dried over anhydrous sodium sulfate, filtered, and evaporated under reduced pressure. The residue was purified by flash chromatography.

5-(1*H*-indol-4-yl)-2-((4-(methyl-sulphonyl)piperazin-1-yl)methyl)-7-(morpholin-4-yl)pyrazolo[1,5-*a*]pyrimidine (**23**)

Compound **23** was prepared from aldehyde **22** (0.39 g, 0.65 mmol), 1-methanesulfonylpiperazine (0.13 g, 0.78 mmol), DCM (4.0 mL) and sodium triacetoxyborohydride (0.25 g, 1.18 mmol). The crude product was purified by flash chromatography (0–10% MeOH gradient in AcOEt) to give **23** (0.27 mg, 0.54 mmol) as a light yellow solid with 84% yield. ^1^HNMR (600 MHz, DMSO-*d*_6_) *δ* 11.31 (s, 1H, NH), 7.64 (dd, *J =* 7.4, 0.8 Hz, 1H, Ar-H), 7.53 (dt, *J =* 8.0, 0.8 Hz, 1H, Ar-H), 7.47 (t, *J =* 2.8 Hz, 1H, Ar-H), 7.22 (t, *J =* 7.7 Hz, 1H, Ar-H), 7.10–7.09 (m, 1H, Ar-H), 6.77 (s, 1H, Ar-H), 6.51 (s, 1H, Ar-H), 3.86–3.84 (m, 4H, morph.), 3.79–3.77 (m, 4H, morph.), 3.74 (s, 2H, CH_2_), 3.14–3.13 (m, 4H), 2.86 (s, 3H, CH_3_), 2.58–2.57 (m, 4H). ^13^CNMR (151 MHz, DMSO-*d*_6_) *δ* 157.9, 153.5, 151.0, 149.5, 136.7, 129.9, 126.4, 125.6, 120.7, 119.5, 113.2, 101.8, 94.7, 91.5, 65.6, 55.5, 51.8, 47.8, 45.4, 33.7. HRMS (ESI): *m*/*z* calcd for C_24_H_29_N_7_O_3_S [M+H]^+^: 496.2125; found 496.2134.

2-((4-(dimethylamino)piperidin-1-yl)methyl)-5-(1*H*-indol-4-yl)-7-(morpholin-4-yl)pyrazolo[1,5-*a*]pyrimidine (**24**)

Compound **24** was prepared from aldehyde **22** (0.18 g, 0.52 mmol), 4-(dimethylamino)piperidine dihydrochloride (0.13 g, 0.62 mmol), DCM (3.5 mL), triethylamine (0.17 mL, 1.24 mmol) and sodium triacetoxyborohydride (0.17 g, 0.78 mmol). The crude product was purified by flash chromatography (0–20% MeOH gradient in AcOEt) to give **24** (0.18 g, 0.39 mmol)) as a light yellow solid with 76% yield. ^1^H NMR (300 MHz, CDCl_3_) *δ*: 8.85 (s; 1H, NH); 7.60 (d; *J =* 7.2 Hz; 1H, Ar-H); 7.51 (d; *J =* 8.2 Hz; 1H, Ar-H); 7.36–7.28 (m; 2H, Ar-H); 7.12–7.08 (m; 1H, Ar-H); 6.65 (s; 1H, Ar-H); 6.61 (s; 1H, Ar-H); 4.04–3.94 (m; 4H, morph.); 3.80 (s; 2H, CH_2_); 3.79–3.72 (m; 4H, morph.); 3.19–3.07 (m; 2H, CH_2_); 2.60–2.49 (m; 1H, CH); 2.44 (s; 6H, 2xCH_3_); 2.22–2.09 (m; 2H, CH_2_); 1.98–1.86 (m; 2H); 1.76–1.59 (m; 2H) HRMS (ESI): *m*/*z* calcd for C_26_H_33_N_7_O [M+H]^+^: 460.2819; found 460.2842.

5-(1*H*-indol-4-yl)-2-((3R)-1-methylpyrrolidin-3-ol)methyl)-7-(morpholin-4-yl)pyrazolo[1,5-*a*]pyrimidine (**25**)

Compound **25** was prepared from aldehyde **22** (0.17 g, 0.48 mmol), (*R*)-(+)-3-pyrrolidinol (53 mg, 0.58 mmol), DCM (2.0 mL) and sodium triacetoxyborohydride (0.18 mg, 0.86 mmol). The crude product was purified by flash chromatography (0–30% MeOH gradient in AcOEt) to give **25** (50 mg, 0.12 mmol) as a light yellow solid with 25% yield. ^1^H NMR (600 MHz, DMSO-*d*_6_) *δ* 11.31 (s, 1H, NH), 7.64 (dd, *J* = 7.4, 0.6 Hz, 1H, Ar-H), 7.53 (d, *J* = 8.0 Hz, 1H, Ar-H), 7.46 (t, *J* = 2.8 Hz, 1H, Ar-H), 7.21 (t, *J* = 7.7 Hz, 1H, Ar-H), 7.09 (t, *J* = 2.1 Hz, 1H, Ar-H), 6.76 (s, 1H, Ar-H), 6.50 (s, 1H, Ar-H), 4.21–4.19 (m, 1H), 3.86–3.84 (m, 4H, morph.), 3.82–3.72 (m, 6H), 2.80–2.77 (m, 1H), 2.71–2.67 (m, 1H), 2.55–2.52 (m, 1H), 2.44–2.42 (m, 1H), 2.01–1.98 (m, 1H), 1.57–1.54 (m, 1H). ^13^C NMR (151 MHz, DMSO-*d*_6_) *δ* 157.8, 154.6, 150.9, 149.5, 136.7, 130.0, 126.4, 125.6, 120.7, 119.5, 113.1, 101.8, 94.6, 91.4, 69.4, 65.6, 62.5, 53.3, 52.3, 47.8, 34.5. HRMS (ESI): *m*/*z* calcd for C_23_H_26_N_6_O_2_ [M+H]^+^: 419.2190; found 419.2191.

5-(1*H*-indol-4-yl)-2-((*3S*)-1-methylpyrrolidin-3-ol)methyl)-7-(morpholin-4-yl)pyrazolo[1,5-*a*]pyrimidine (**26**)

Compound **26** was prepared from aldehyde **22** (0.17 mg, 0.48 mmol), (*S*)-3-pyrrolidinol (52 mg, 0.58 mmol), DCM (2.0 mL) and sodium triacetoxyborohydride (0.18 mg, 0.86 mmol). The crude product was purified by flash chromatography (0–30% MeOH gradient in AcOEt) to give **26** (70 mg, 0.17 mmol) as a light yellow solid with 35% yield. ^1^H NMR (600 MHz, DMSO-*d*_6_) *δ* 11.33 (s, 1H, NH), 7.64 (dd, *J* = 7.4, 0.7 Hz, 1H, Ar-H), 7.53 (d, *J* = 8.0 Hz, 1H, Ar-H), 7.47 (t*, J* = 2.8 Hz, 1H, Ar-H), 7.22–7.20 (m, 1H, Ar-H), 7.09–7.08 (m, 1H, Ar-H), 6.76 (s, 1H, Ar-H), 6.50 (s, 1H, Ar-H), 4.21–4.19 (m, 1H), 3.86–3.84 (m, 4H, morph.), 3.78–3.72 (m, 6H), 2.78 (m, 1H), 2.68 (m, 1H), 2.54–2.50 (m, 1H), 2.43 (m, 1H), 2.03–1.97 (m, 1H), 1.57–1.54 (m, 1H). ^13^C NMR (151 MHz, DMSO-*d*_6_) *δ* 157.8, 154.6, 150.9, 149.6, 136.8, 130.0, 126.5, 125.6, 120.8, 119.6, 113.2, 101.8, 94.6, 91.5, 69.5, 65.6, 62.6, 53.4, 52.4, 47.9, 34.5. HRMS (ESI): *m*/*z* calcd for C_23_H_26_N_6_O_2_ [M+H]^+^: 419.2190; found 419.2200.

2-((1,1-dioxothiomorpholin-1-yl)-methyl)-5-(1*H*-indol-4-yl)-7-(morpholin-4-yl)pyrazolo[1,5-*a*]pyrimidine (**27**)

Compound **27** was prepared from aldehyde **22** (85 mg, 0.25 mmol), thiomorpholine-1,1-dioxide (40 mg, 0.29 mmol), DCM (3.0 mL) and sodium triacetoxyborohydride (78 mg, 0.37 mmol). The crude product was purified by flash chromatography (0–10% MeOH gradient in AcOEt) to give **27** (48 mg, 0.10 mmol) as a light yellow solid with 42% yield. ^1^H NMR (400 MHz, DMSO-*d*_6_) *δ* 11.33 (s, 1H, NH), 7.62 (dd, *J* = 7.4, 0.8 Hz, 1H, Ar-H), 7.51 (d, *J* = 8.1 Hz, 1H, Ar-H), 7.44 (t, *J* = 2.8 Hz, 1H, Ar-H), 7.19 (t, *J* = 7.7 Hz, 1H, Ar-H), 7.07–7.06 (m, 1H, Ar-H), 6.75 (s, 1H, Ar-H), 6.54 (s, 1H, Ar-H), 3.87 (s, 2H), 3.83–3.81 (m, 3H), 3.75–3.74 (m, 3H), 3.12–3.09 (m, 4H), 2.98–2.95 (m, 4H). ^13^C NMR (101 MHz, DMSO-*d*_6_) *δ* 158.0, 153.3, 151.0, 149.6, 136.8, 129.9, 126.5, 125.6, 120.8, 119.6, 113.3, 101.8, 94.8, 91.7, 65.6, 54.2, 50.6, 50.2, 47.9. HRMS (ESI): *m*/*z* calcd for C_23_H_26_N_6_O_3_S [M+H]^+^: 467.1860; found 467.1866.

5-(1*H*-indol-4-yl)-7-(morpholin-4-yl)-2-((morpholin-4-yl)methyl)pyrazolo[1,5-*a*]pyrimidine (**28**)

Compound **28** was prepared from aldehyde **22** (0.20 g, 0.23 mmol), morpholine (24 mL, 24 mg, 0.27 mmol), DCM (3.0 mL) and sodium triacetoxyborohydride (95 mg, 0.45 mmol). The crude product was purified by flash chromatography (0–10% MeOH gradient in AcOEt) to give **28** (55 mg, 0.13 mmol) as a light yellow solid with 59% yield. ^1^H NMR (500 MHz, DMSO-*d*_6_) *δ* 11.32 (s, 1H, NH), 7.65 (d, *J* = 7.4 Hz, 1H, Ar-H), 7.54 (d*, J* = 7.4 Hz, 1H, Ar-H), 7.49–7.45 (m, 1H, Ar-H), 7.25–7.19 (m, 1H, Ar-H), 7.12–7.08 (m, 1H, Ar-H), 6.77 (s, 1H, Ar-H), 6.52 (s, 1H, Ar-H), 3.89–3.83 (m, 4H, morph.), 3.81–3.76 (m, 4H, morph.), 3.68 (s, 2H, CH_2_), 3.63–3.57 (m, 4H, morph.), 2.49–2.45 (m, 4H, morph.). ^13^C NMR (126 MHz, DMSO-*d*_6_) *δ*: 157.9, 153.6, 151.0, 149.6, 136.8, 129.9, 126.5, 125.6, 120.8, 119.6, 113.2, 101.8, 94.8, 91.5, 66.2, 65.6, 56.4, 53.2, 47.9. HRMS (ESI): *m*/*z* calcd for C_23_H_26_N_6_O_2_ [M+H]^+^: 419.2190; found 419.2196.

5-(1*H*-indol-4-yl)-2-((4-(4-methylpiperazin-1-yl)piperidin-1-yl)methyl)-7-(morpholin-4-yl)pyrazolo[1,5-*a*]pyrimidine (**29**)

Compound **29** was prepared from aldehyde **22** (0.17 g, 0.50 mmol), 1-methyl-4-(piperidin-4-yl)piperazine (0.11 g, 0.6 mmol), DCM (3.5 mL) and sodium triacetoxyborohydride (0.16 g, 0.75 mmol). The crude product was purified by flash chromatography (0–15% MeOH gradient in AcOEt) to give **29** (0.23 g, 0.45 mmol) as a yellow solid with 89% yield. ^1^H NMR (300 MHz, CDCl_3_) *δ*: 9.56 (s; 1H); 7.62–7.55 (m; 1H, Ar-H); 7.47–7.1 (m; 1H, Ar-H); 7.31–7.21 (m; 2H, Ar-H); 7.11–7.02 (m; 1H, Ar-H); 6.64 (s; 1H, Ar-H); 6.62 (s; 1H, Ar-H); 4.00–3.90 (m; 4H, morph.); 3.86–3.62 (m; 6H); 3.19–3.05 (m; 2H); 2.81–2.45 (m; 8H); 2.34 (s; 3H); 2.39–2.29 (m; 1H); 2.21–2.08 (m; 2H); 1.91–1.79 (m; 2H); 1.75–1.54 (m; 2H). ^13^C NMR (75 MHz, CDCl_3_) *δ* 158.8, 154.2, 151.6, 150.2, 136.8, 131.1, 126.1, 125.8, 121.8, 120.1, 113.0, 102.4, 96.0, 92.5, 66.3, 61.9, 56.5, 54.8, 53.0, 48.5, 48.4, 45.6, 27.9. HRMS (ESI): *m*/*z* calcd for C_29_H_38_N_8_O [M+H]^+^: 515.3241; found 515.3224.

3-ethyl-1-(1-((5-(1H-indol-4-yl)-7-(morpholin-4-yl)pyrazolo[1,5-a]pyrimidin-2-yl)methyl)piperidin-4-yl)urea (**30**)

The 3-ethyl-1-(piperidin-4-yl)urea was synthesizedaccording to the van Duzer et al. procedure [56]. The urea derivative was used in the reductive amination reaction (next step) as is, without additional purification.

Compound **30** was prepared from aldehyde **22** (0.20 g, 0.58 mmol), 3-ethyl-1-(piperidin-4-yl)urea hydrochloride (0.14 g, 0.69 mmol), DCM (4.0 mL), triethylamine (0.194 mL, 1.38 mmol) and sodium triacetoxyborohydride (0.19 g, 0.86 mmol). The crude product was purified by flash chromatography (0–15% MeOH gradient in AcOEt) to give **30** (0.15 g, 0.30 mmol) as a light yellow solid with 52% yield. ^1^H NMR (400 MHz, DMSO-*d*_6_) *δ* 11.33 (s, 1H), 7.64 (d, *J* = 7.4 Hz, 1H), 7.53 (d, *J* = 8.0 Hz, 1H), 7.47 (t, *J* = 2.7 Hz, 1H), 7.21 (t, *J* = 7.7 Hz, 1H, Ar-H), 7.09–7.09 (m, 1H, Ar-H), 6.76 (s, 1H, Ar-H), 6.48 (s, 1H, Ar-H), 5.71 (d, *J* = 7.8 Hz, 1H), 5.65 (t, *J* = 5.5 Hz, 1H), 3.86–3.84 (m, 4H), 3.78–3.77 (m, 4H), 3.64 (s, 2H), 3.36–3.36 (m, 1H), 3.01–2.94 (m, 2H), 2.81–2.78 (m, 2H), 2.15–2.10 (m, 2H), 1.74–1.72 (m, 2H, CH_2_), 1.38–1.32 (m, 2H, CH_2_), 0.96 (t, *J* = 7.1 Hz, 3H, CH_3_). ^13^C NMR (101 MHz, DMSO-*d*_6_) *δ* 157.9, 157.3, 154.3, 151.0, 149.5, 136.8, 130.0, 126.5, 125.6, 120.8, 119.6, 113.2, 101.8, 94.7, 91.5, 65.6, 56.2, 52.0, 47.9, 46.1, 33.9, 32.5, 15.7. HRMS (ESI): *m*/*z* calcd for C_27_H_34_N_8_O_2_ [M+H]^+^: 503.2877; found 503.2882.

1-phenyl-3-(1-((5-(1H-indol-4-yl)-7-(morpholin-4-yl)pyrazolo[1,5-a]pyrimidin-2-yl)methyl)piperidin-4-yl)urea (**31**)

The synthesis of 1-phenyl-3-(piperidin-4-yl)urea was conducted according to the van Duzer et al.procedure [56]. The urea derivative was used in the reductive amination reaction (next step)as is, without additional purification.

Compound **31** was prepared from aldehyde **22** (0.20 g, 0.58 mmol), 1-phenyl-3-(piperidin-4-yl)urea hydrochloride (0.18 g, 0.69 mmol), DCM (4.0 mL), triethylamine (0.194 mL, 1.38 mmol) and sodium triacetoxyborohydride (0.19 g, 0.86 mmol). The crude product was purified by flash chromatography (0–15% MeOH gradient in AcOEt) to give **31** (0.18 g, 0.33 mmol) as a light yellow solid with 58% yield. ^1^H NMR (400 MHz, DMSO-*d*_6_) *δ* 11.33 (s, 1H, NH), 8.30 (s, 1H), 7.65 (dd, *J* = 7.4, 0.8 Hz, 1H), 7.53 (d, *J* = 8.1 Hz, 1H), 7.47 (t, *J* = 2.8 Hz, 1H), 7.37–7.34 (m, 2H), 7.24–7.16 (m, 3H), 7.10–7.09 (m, 1H, Ar-H), 6.88–6.84 (m, 1H, Ar-H), 6.76 (s, 1H, Ar-H), 6.51 (s, 1H, Ar-H), 6.11 (d, *J* = 7.7 Hz, 1H), 3.86–3.84 (m, 4H, morph.), 3.79–3.78 (m, 4H, morph.), 3.67 (s, 2H, CH_2_), 3.49–3.48 (m, 1H), 2.83–2.80 (m, 2H, CH_2_), 2.22–2.17 (m, 2H, CH_2_), 1.98–1.80 (m, 2H, CH_2_), 1.47–1.38 (m, 2H, CH_2_). ^13^C NMR (101 MHz, DMSO-*d*_6_) *δ* 157.9, 154.5, 154.2, 151.0, 149.6, 140.5, 136.8, 130.0, 128.6, 126.5, 125.6, 120.9, 120.8, 119.6, 117.5, 113.2, 101.8, 94.7, 91.5, 65.6, 56.2, 51.7, 47.9, 46.0, 32.2. HRMS (ESI): *m*/*z* calcd for C_31_H_34_N_8_O_2_ [M+H]^+^: 551.2887; found 551.2880.

5-(1*H*-indol-4-yl)-2-((4-(4-methoxyphenyl)piperazin-1-yl)-methyl)-7-(morpholin-4-yl)pyrazolo[1,5-*a*]pyrimidine (**32**)

Compound **32** was prepared from aldehyde **22** (0.18 g, 0.52 mmol), 1-(4-methoxyphenyl)piperazine (0.12 g, 0.63 mmol), DCM (3.5 mL) and sodium triacetoxyborohydride (0.17 g, 0.79 mmol). The crude product was purified by flash chromatography (0–5% MeOH gradient in AcOEt) to give **32** (0.22 g, 0.42 mmol) as a light yellow solid with 81% yield. ^1^H NMR (300 MHz, CDCl_3_) *δ*: 8.52 (s; 1H, NH); 7.61 (d; *J* = 7.4 Hz; 1H, Ar-H); 7.49 (d; *J* = 8.1 Hz; 1H, Ar-H); 7.35–7.27 (m; 2H, Ar-H); 7.14–7.09 (m; 1H, Ar-H); 6.96–6.80 (m; 4H); 6.67 (s; 1H, Ar-H); 6.65 (s; 1H, Ar-H); 4.04–3.95 (m; 4H, morph.); 3.88 (s; 2H, CH_2_); 3.82–3.76 (m; 4H, morph.); 3.77 (s; 3H, CH_3_); 3.19–3.11 (m; 4H, piperaz.); 2.84–2.74 (m; 4H, piperaz.). ^13^C NMR (75 MHz, CDCl_3_) *δ* 158.8, 154.3, 153.8, 151.7, 150.3, 145.8, 136.8, 131.3, 126.1, 125.6, 122.0, 120.3, 118.3, 114.5, 112.9, 102.7, 96.1, 92.5, 66.4, 56.8, 55.7, 53.4, 50.7, 48.6. HRMS (ESI): *m*/*z* calcd for C_30_H_33_N_7_O_2_ [M+H]^+^: 524.2769; found 524.2770.

5-(1*H*-indol-4-yl)-2-((4-methyl-piperazin-1-yl)methyl)-7-(morpholin-4-yl)pyrazolo[1,5-*a*]pyrimidine (**33**)

Compound **33** was prepared from aldehyde **22** (85 mg, 0.24 mmol), 1-methylpiperazine, (33 mL, 29 mg, 0.29 mmol), DCM (4.0 mL) and sodium triacetoxyborohydride (78 mg, 0.37 mmol). The crude product was purified by flash chromatography (0–20% MeOH gradient in AcOEt) to give **33** (91 mg, 0.21 mmol) as a light yellow solid with 86% yield. ^1^H NMR (400 MHz, DMSO-*d*_6_) *δ* 11.41 (s, 1H, NH), 7.64 (d, *J* = 7.4 Hz, 1H, Ar-H), 7.53 (d, *J* = 8.1 Hz, 1H, Ar-H), 7.46 (t, *J* = 2.7 Hz, 1H, Ar-H), 7.21 (t, *J* = 7.7 Hz, 1H, Ar-H), 7.09 (t*, J* = 2.0 Hz, 1H, Ar-H), 6.76 (s, 1H, Ar-H), 6.48 (s, 1H, Ar-H), 3.85–3.83 (m, 4H, morph.), 3.78–3.76 (m, 4H, morph.), 3.65 (s, 2H, CH_2_), 2.50–2.45 (m, 4H, piperaz.), 2.32–2.32 (m, 4H, piperaz.), 2.14 (s, 3H, CH_3_). ^13^C NMR (101 MHz, DMSO-*d*_6_) *δ* 157.9, 154.0, 151.0, 149.6, 136.8, 129.9, 126.5, 125.6, 120.7, 119.6, 113.2, 101.8, 94.7, 91.5, 65.6, 56.0, 54.7, 52.6, 47.9, 45.7. HRMS (ESI): *m*/*z* calcd for C_24_H_29_N_7_O [M+H]^+^: 432.2506; found 432.2511.

2-((4-ethylpiperazin-1-yl)methyl)-5-(1*H*-indol-4-yl)-7-(morpholin-4-yl)pyrazolo[1,5-*a*]pyrimidine (**34**)

Compound **34** was prepared from aldehyde **22** (85 mg, 0.24 mmol), 1-ethylpiperazine (37 mL, 33 mg, 0.29 mmol), DCM (4.0 mL) and sodium triacetoxyborohydride (78 mg, 0.37 mmol). The crude product was purified by flash chromatography (0–20% MeOH gradient in AcOEt) to give **34** (85 mg, 0.19 mmol) as a light yellow solid with 78% yield. ^1^H NMR (400 MHz, DMSO-*d*_6_) *δ* 11.37 (s, 1H, NH), 7.64 (d, *J* = 7.4 Hz, 1H, Ar-H), 7.53 (d, *J* = 8.1 Hz, 1H, Ar-H), 7.46 (t, *J* = 2.7 Hz, 1H, Ar-H), 7.21 (t, *J* = 7.7 Hz, 1H, Ar-H), 7.09–7.08 (m, 1H, Ar-H), 6.76 (s, 1H, Ar-H), 6.49 (s, 1H, Ar-H), 3.86–3.84 (m, 4H, morph.), 3.78–3.77 (m, 4H, morph.), 3.67 (s, 2H, CH_2_), 2.63–2.44 (m, 8H, piperaz.), 2.39 (q, *J* = 7.2 Hz, 2H, CH_2_), 1.00 (t, *J* = 7.2 Hz, 3H, CH_3_). ^13^C NMR (101 MHz, DMSO-*d*_6_) *δ* 172.0, 157.9, 153.8, 151.0, 149.6, 136.8, 129.9, 126.5, 125.6, 120.7, 119.6, 113.2, 101.8, 94.8, 91.5, 65.6, 55.9, 52.1, 51.4, 47.9, 21.1. HRMS (ESI): *m*/*z* calcd for C_25_H_31_N_7_O [M+H]^+^: 446.2663; found 446.2661.

Methyl 1-((5-(1*H*-indol-4-yl)-7-(morpholin-4-yl)pyrazolo[1,5-*a*]pyrimidin-2-yl)methyl)piperidin-4-carboxylate (**35**)

Compound **35** was prepared from aldehyde **22** (85 mg, 0.24 mmol), methyl isonipecotate, (42 mg, 0.29 mmol), DCM (4.0 mL) and sodium triacetoxyborohydride (78 mg, 0.37 mmol). The crude product was purified by flash chromatography (0–10% MeOH gradient in CHCl_3_) to give **35** (76 mg, 0.16 mmol) as a light yellow solid with 65% yield. ^1^H NMR (400 MHz, DMSO-*d*_6_) *δ* 11.35 (s, 1H, NH), 7.64 (dd, *J* = 7.4, 0.7 Hz, 1H, Ar-H), 7.53 (d, *J* = 8.1 Hz, 1H, Ar-H), 7.47 (t, *J* = 2.8 Hz, 1H, Ar-H), 7.21 (t, *J* = 7.7 Hz, 1H, Ar-H), 7.10–7.09 (m, 1H, Ar-H), 6.76 (s, 1H, Ar-H), 6.49 (s, 1H, Ar-H), 3.86–3.83 (m, 4H, morhp.), 3.78–3.77 (m, 4H, morph.), 3.65 (s, 2H, CH_2_), 3.58 (s, 3H, CH_3_), 2.87–2.84 (m, 2H, CH_2_), 2.32–2.27 (m, 1H, CH), 2.11–2.06 (m, 2H, CH_2_), 1.82–1.79 (m, 2H, CH_2_), 1.63–1.57 (m, 2H). ^13^C NMR (101 MHz, DMSO-*d*_6_) *δ* 174.8, 157.9, 154.0, 151.0, 149.5, 136.8, 130.0, 126.5, 125.6, 120.7, 119.6, 113.2, 101.8, 94.7, 91.5, 65.6, 56.2, 52.2, 51.3, 47.9, 40.1, 28.0. HRMS (ESI): *m*/*z* calcd for C_26_H_30_N_6_O_3_ [M+H]^+^: 475.2452; found 475.2458.

2-((4-(2-hydroxypropan-2-yl)piperidin-1-yl)methyl)-5-(1*H*-indol-4-yl)-7-(morpholin-4-yl)pyrazolo[1,5-*a*]pyrimidine (**36**)

Compound **36** was prepared from aldehyde **22** (0.18 g, 0.52 mmol), 2-(4-piperidyl)-2-propanol (93 mg, 0.62 mmol), DCM (3.5 mL) and sodium triacetoxyborohydride (0.17 g, 0.78 mmol). The crude product was purified by flash chromatography (0–5% MeOH gradient in AcOEt) to give **36** (0.183 g, 0.39 mmol) as an off-white solid with 74% yield. ^1^H NMR (300 MHz, CDCl_3_) *δ*: 8.57 (s; 1H, NH); 7.64–7.58 (m; 1H, Ar-H); 7.52–7.46 (m; 1H, Ar-H); 7.36–7.25 (m; 2H, Ar-H); 7.14–7.09 (m; 1H, Ar-H); 6.64 (s; 1H, Ar-H); 6.64 (s; 1H, Ar-H); 4.03–3.95 (m; 4H, morph.); 3.82 (s; 2H, CH_2_); 3.81–3.71 (m; 4H, morph.); 3.20–3.10 (m; 2H, CH_2_); 2.18–2.03 (m; 2H, CH_2_); 1.81–1.69 (m; 2H, CH_2_); 1.56–1.38 (m; 2H, CH_2_); 1.35–1.30 (m; 1H); 1.18 (s; 6H). ^13^C NMR (75 MHz, CDCl_3_) *δ* 158.7, 151.7, 150.2, 136.8, 131.3, 126.1, 125.6, 122.0, 120.3, 112.9, 102.7, 96.2, 92.4, 72.7, 66.4, 56.9, 54.2, 48.6, 47.3, 27.1, 27.0. HRMS (ESI): *m*/*z* calcd for C_27_H_34_N_6_O_6_ [M+H]^+^: 475.2816; found 475.2815.

2-((4-*tert*-butylpiperazin-1-yl)methyl)-5-(1*H*-indol-4-yl)-7-(morpholin-4-yl)pyrazolo[1,5-*a*]pyrimidine (**37**)

Compound **37** was prepared from aldehyde **22** (0.12 g, 0.35 mmol), *N*-*tert*-butylpiperazine (59 mg, 0.42 mmol), DCM (2.0 mL) and sodium triacetoxyborohydride (0.11 g, 0.52 mmol). The crude product was purified by flash chromatography (0–20% MeOH gradient in AcOEt) to give **37** (0.15 g, 0.32 mmol) as a yellow solid with 93% yield. ^1^H NMR (400 MHz, DMSO-*d*_6_) *δ* 11.33 (s, 1H, NH), 7.64 (dd, *J* = 7.4, 0.9 Hz, 1H, Ar-H), 7.53 (dt, *J* = 8.1, 0.8 Hz, 1H, Ar-H), 7.46 (t, *J* = 2.8 Hz, 1H, Ar-H), 7.21 (t, *J* = 7.7 Hz, 1H, Ar-H), 7.10–7.08 (m, 1H, Ar-H), 6.76 (s, 1H, Ar-H), 6.48 (s, 1H, Ar-H), 3.86–3.84 (m, 4H, morph.), 3.78–3.76 (m, 4H, morph.), 3.63 (s, 2H, CH_2_), 2.53–2.45 (m, 8H, piperaz.), 0.98 (s, 9H, *t*-Bu.). ^13^C NMR (101 MHz, DMSO-*d*_6_) *δ* 157.8, 154.0, 151.0, 149.5, 136.8, 130.0, 126.5, 125.6, 120.7, 119.6, 113.2, 101.8, 94.8, 91.5, 65.6, 56.0, 53.5, 53.1, 47.9, 45.2, 25.7. HRMS (ESI): *m*/*z* calcd for C_27_H_35_N_7_O [M+H]^+^: 474.2976; found 474.2976.

2-(4-((5-(1*H*-indol-4-yl)-7-(morpholin-4-yl)pyrazolo[1,5-*a*]pyrimidin-2-yl)methyl)piperazin-1-yl)-2-methylpropionamide (**38**)

Compound **38** was prepared from aldehyde **22** (0.20 g, 0.58 mmol), 2-methyl-2-(piperazin-1-yl)propenamide dihydrochloride (0.18 g, 0.69 mmol), DCM (3.0 mL), triethylamine (0.194 mL, 1.38 mmol) and sodium triacetoxyborohydride (0.18 g, 0.86 mmol). The crude product was purified by flash chromatography (0–10% MeOH gradient in AcOEt) to give **38** (0.21 g, 0.42 mmol) as a light yellow solid with 73% yield. ^1^H NMR (500 MHz, DMSO-*d*_6_) *δ* 11.32 (s, 1H, NH), 7.65 (d, *J* = 7.3 Hz, 1H, Ar-H), 7.54 (d, *J* = 8.0 Hz, 1H, Ar-H), 7.49–7.45 (m, 1H, Ar-H), 7.22 (t, *J* = 7.7 Hz, 1H, Ar-H), 7.12–7.08 (m, 1H, Ar-H), 7.07–7.01 (m, 1H, Ar-H), 6.95–6.90 (m, 1H), 6.77 (s, 1H), 6.50 (s, 1H), 3.89–3.82 (m, 4H, morph.), 3.82–3.76 (m, 4H, morph.), 3.68 (s, 2H, CH_2_), 2.57–2.51 (m, 4H), 2.49–2.40 (m, 4H), 1.06 (s, 6H). ^13^C NMR (126 MHz, DMSO-*d*_6_) *δ* 178.1, 157.9, 153.9, 151.0, 149.5, 136.8, 130.0, 126.5, 125.6, 120.8, 119.6, 113.2, 101.8, 94.8, 91.5, 65.6, 62.4, 56.0, 53.2, 47.9, 46.1, 20.4. HRMS (ESI): *m*/*z* calcd for C_27_H_34_N_8_O_2_ [M+H]^+^: 503.2877; found 503.2901.

2-((4-cyclopropylpiperazin-1-yl)methyl)-5-(1*H*-indol-4-yl)-7-(morpholin-4-yl)pyrazolo[1,5-*a*]pyrimidine (**39**)

Compound **39** was prepared from aldehyde **22** (85 mg, 0.25 mmol), 1-cyclopropylpiperazine (35 mL, 37 mg, 0.29 mmol), DCM (3.0 mL) and sodium triacetoxyborohydride (78 mg, 0.37 mmol). The crude product was purified by flash chromatography (0–15% MeOH gradient in AcOEt) to give **39** (84 mg, 0.18 mmol) as a light yellow solid with 75% yield. ^1^H NMR (400 MHz, DMSO-*d*_6_) *δ* 11.32 (s, 1H, NH), 7.64 (dd*, J* = 7.4, 0.7 Hz, 1H, Ar-H), 7.53 (d, *J* = 8.1 Hz, 1H, Ar-H), 7.47 (t, *J* = 2.8 Hz, 1H, Ar-H), 7.21 (t, *J* = 7.7 Hz, 1H, Ar-H), 7.09 (t, *J* = 2.1 Hz, 1H), 6.76 (s, 1H, Ar-H), 6.48 (s, 1H, Ar-H), 3.85–3.83 (m, 4H, morph.), 3.78–3.76 (m, 4H, morph.), 3.64 (s, 2H, CH_2_), 2.54 (s, 4H, piperaz.), 2.43 (s, 4H, piperaz.), 1.58 (s, 1H, CH), 0.38–0.36 (m, 2H, CH_2_), 0.26–0.24 (m, 2H, CH_2_). ^13^C NMR (101 MHz, DMSO-*d*_6_) *δ* 157.9, 154.0, 151.0, 149.5, 136.8, 130.0, 126.5, 125.6, 120.7, 119.6, 113.2, 101.8, 94.7, 91.5, 65.6, 56.0, 52.7, 52.7, 47.9, 38.0, 5.6. HRMS (ESI): *m*/*z* calcd for C_26_H_31_N_7_O [M+H]^+^: 458.2663; found 458.2666.

2-((4-cyclopentylpiperazin-1-yl)methyl)-5-(1*H*-indol-4-yl)-7-(morpholin-4-yl)pyrazolo[1,5-*a*]pyrimidine (**40**)

Compound **40** was prepared from aldehyde **22** (70 mg, 0.20 mmol), 1-cyclopentylpiperazine (39 mL, 38 mg, 0.24 mmol), DCM (4.0 mL) and sodium triacetoxyborohydride (64 mg, 0.30 mmol). The crude product was purified by flash chromatography (0–15% MeOH gradient in AcOEt) to give **40** (84 mg, 0.17 mmol) as a light yellow solid with 86% yield. ^1^H NMR (400 MHz, DMSO-*d*_6_) *δ* 11.34 (s, 1H, NH), 7.64 (dd, *J* = 7.4, 0.7 Hz, 1H, Ar-H), 7.53 (d, *J* = 8.1 Hz, 1H, Ar-H), 7.46 (t*, J* = 2.8 Hz, 1H, Ar-H), 7.21 (t, *J* = 7.7 Hz, 1H, Ar-H), 7.09 (t, *J* = 2.1 Hz, 1H, Ar-H), 6.76 (s, 1H, Ar-H), 6.48 (s, 1H, Ar-H), 3.86–3.83 (m, 4H, morph.), 3.78–3.76 (m, 4H, morph.), 3.64 (s, 2H, CH_2_), 2.53–2.42 (m, 9H), 1.75–1.72 (m, 2H, CH_2_), 1.59–1.55 (m, 2H), 1.48–1.44 (m, 2H), 1.31–1.26 (m, 2H). ^13^C NMR (101 MHz, DMSO-*d*_6_) *δ* 157.9, 153.9, 151.0, 149.5, 136.8, 130.0, 126.4, 125.6, 120.7, 119.6, 113.2, 101.8, 94.7, 91.5, 66.7, 65.6, 56.0, 52.7, 51.6, 47.9, 29.8, 23.6. HRMS (ESI): *m*/*z* calcd for C_28_H_35_N_7_O [M+H]^+^: 486.2976; found 486.2973.

2-((4-tert-butylpiperidin-1-yl)methyl)-5-(1*H*-indol-4-yl)-7-(morpholin-4-yl)pyrazolo[1,5-*a*]pyrimidine (**41**)

Compound **41** was prepared from aldehyde **22** (0.10 g, 0.29 mmol), 4-(*tert*-butyl)piperidine hydrochloride (61 mg, 0.35 mmol), DCM (4.0 mL), triethylamine (0.097 mL, 0.69 mmol) and sodium triacetoxyborohydride (94 mg, 0.43 mmol). The crude product was purified by flash chromatography (0–20% MeOH gradient in CHCl_3_) to give **41** (0.10 g, 0.21 mmol) as a light yellow solid with 76% yield. ^1^H NMR (400 MHz, DMSO-*d*_6_) 11.32 (s, 1H, NH), 7.64 (dd, *J* = 7.4, 0.7 Hz, 1H, Ar-H), 7.53 (d, *J* = 8.1 Hz, 1H, Ar-H), 7.46 (t, *J* = 2.8 Hz, 1H, Ar-H), 7.21 (t, *J* = 7.7 Hz, 1H, Ar-H), 7.10–7.09 (m, 1H, Ar-H), 6.75 (s, 1H, Ar-H), 6.48 (s, 1H, Ar-H), 3.85–3.83 (m, 4H, morph.), 3.78–3.76 (m, 4H, morph.), 3.61 (s, 2H, CH_2_), 2.98–2.95 (m, 2H, CH_2_), 1.95–1.89 (m, 2H, CH_2_), 1.59–1.56 (m, 2H, CH_2_), 1.27–1.17 (m, 2H, CH_2_), 0.96–0.90 (m, 1H, CH), 0.81 (s, 9H, *t*-Bu.). ^13^C NMR (101 MHz, DMSO-*d*_6_) *δ* 157.8, 154.3, 151.0, 149.5, 136.8, 130.0, 126.4, 125.6, 120.7, 119.5, 113.2, 101.9, 94.7, 91.4, 65.6, 56.4, 54.0, 47.9, 45.8, 31.8, 27.2, 26.4. HRMS (ESI): *m*/*z* calcd for C_28_H_36_N_6_O [M+H]^+^: 473.3023; found 473.3028.

5-(1*H*-indol-4-yl)-2-((4-(oxetan-3-yl)piperidin-1-yl)methyl)-7-(morpholin-4-yl)pyrazolo[1,5-*a*]pyrimidine (**42**)

The multistep preparation of compound **42** started from 1-(oxetan-3-yl) piperazine.

Step 1.

To the solution of 3-oxetanone (0.23 mL, 0.28 g, 3.9 mmol) in dry DCM (39.0 mL), 1-Boc-piperazine (0.60 g, 3.2 mmol) was added, and then the mixture was stirred at room temperature. After four h, sodium triacetoxyborohydride (1.35 g, 6.4 mmol) was added, and stirring was continued at room temperature overnight. Then, water (30 mL) was added to the reaction mixture, and the phases were separated. The aqueous phase was extracted three times with chloroform (25 mL). Combined organic phases were dried over anhydrous sodium sulfate, filtrated the drying agent, and the solvent was evaporated under reduced pressure to obtain *tert*-butyl 4-(oxetan-3-yl)piperazin-1-carboxylate (0.61 g, 2.52 mmol) with 65% yield without purification. ^1^H NMR (300 MHz, CDC1_3_) *δ* 4.68–4.52 (m; 4H, piperaz.); 3.50–3.32 (m; 5H); 2.31–2.09 (m; 4H); 1.43 (s; 9H, *t*-Bu.). MS-ESI: (*m*/*z*) calcd for C_12_H_22_N_2_O_3_ [M+H]^+^: 243.17; found 243.2.

Step 2.

To the solution of the product of Step 1 (0.55 g, 2.8 mmol) in DCM (28 mL), trifluoroacetic acid (16.8 mL) was added. The reaction was carried out at room temperature for two h. Then, the water was added (30 mL), and the reaction mixture was alkalized with saturated sodium carbonate solution (10 mL). Phases were separated, and the aqueous phase was extracted three times with chloroform (25 mL). Combined organic phases were dried over anhydrous sodium sulfate. The drying agent was filtered off and the solvent evaporated under reduced pressure to obtain 1-(oxetan-3-yl)piperazine (0.23 g, 1.61 mmol) with 57% yield without purification. ^1^H NMR (300 MHz, CDCl_3_) *δ* 4.66–4.56 (m; 4H); 3.66–3.56 (m; 1H); 3.30–3.12 (m; 4H); 2.68–2.51 (m; 4H). MS-ESI: (*m*/*z*) calcd for C_7_H_14_N_2_O [M+H]^+^: 143.12; found 143.1.

Compound **42** was prepared from aldehyde **22** (0.20 g, 0.58 mmol), 1-(oxetan-3-yl)piperazine (98 mg, 0.69 mmol), DCM (4.0 mL) and sodium triacetoxyborohydride (0.19 g, 0.86 mmol). The crude product was purified by flash chromatography (0–10% MeOH gradient in AcOEt) to give **42** (0.15 g, 0.32 mmol) as a light yellow solid with 54% yield. ^1^H NMR (400 MHz, DMSO-*d*_6_) *δ* 11.33 (s, 1H, NH), 7.64 (dd, *J* = 7.5, 0.8 Hz, 1H, Ar-H), 7.54–7.52 (m, 1H, Ar-H), 7.47 (t, *J* = 2.8 Hz, 1H, Ar-H), 7.21 (t, *J* = 7.7 Hz, 1H, Ar-H), 7.10–7.08 (m, 1H, Ar-H), 6.76 (s, 1H, Ar-H), 6.49 (s, 1H, Ar-H), 4.50 (t, *J* = 6.5 Hz, 2H, CH_2_), 4.39 (t, *J* = 6.1 Hz, 2H, CH_2_), 3.86–3.84 (m, 4H, morph.), 3.78–3.76 (m, 4H, morph.), 3.67 (s, 2H, CH_2_), 3.40–3.33 (m, 1H, CH), 2.53–2.48 (m, 4H, piperaz.), 2.27–2.27 (m, 4H, piperaz.). ^13^C NMR (101 MHz, DMSO-*d*_6_) *δ* 157.9, 153.9, 151.0, 149.5, 136.8, 130.0, 126.5, 125.6, 120.7, 119.6, 113.2, 101.8, 94.8, 91.5, 74.4, 65.6, 58.5, 56.0, 52.3, 49.0, 47.9. HRMS (ESI): *m*/*z* calcd for C_26_H_31_N_7_O_2_ [M+H]^+^: 474.2612; found 474.2616.

5-(1*H*-indol-4-yl)-2-(((1*S*, 4*S*)-2-(oxetan-3-yl)-2,5-diaza-bicyclo [2.2.1]hept-2-yl)methyl)-7-(morpholin-4-yl)pyrazolo[1,5-*a*]pyrimidine (**43**)

The preparation of compound **43** started from (*1S, 4S*)-2-(oxetan-3-yl)-2,5-diazabicyclo [2.2.1]heptane, which was prepared analogously, as described for the synthesis of 1-(oxetan-3-yl) piperazine.

Compound **43** was prepared from aldehyde **22** (0.20 g, 0.58 mmol), (*1S, 4S*)-2-(oxetan-3-yl)-2,5-diazabicyclo [2.2.1]heptane (0.11 g, 0.69 mmol), DCM (4.0 mL) and sodium triacetoxyborohydride (0.19 g, 0.86 mmol). The crude product was purified by flash chromatography (0–10% MeOH gradient in AcOEt) to give **43** (0.18 g, 0.37 mmol) as a light yellow solid with 63% yield. ^1^H NMR (600 MHz, DMSO-*d*_6_) *δ* 11.34 (s, 1H, NH), 7.63 (dd, *J* = 7.4, 0.7 Hz, 1H, Ar-H), 7.53 (d, J = 8.0 Hz, 1H, Ar-H), 7.46 (t, *J* = 2.8 Hz, 1H, Ar-H), 7.21 (t, *J* = 7.7 Hz, 1H, Ar-H), 7.09–7.08 (m, 1H, Ar-H), 6.75 (s, 1H, Ar-H), 6.50 (s, 1H, Ar-H), 4.59–4.51 (m, 2H, CH_2_), 4.41–4.34 (m, 2H, CH_2_), 3.89–3.82 (m, 6H), 3.79–3.75 (m, 5H), 3.40 (s, 1H, CH), 3.22 (s, 1H), 2.84 (d, *J* = 9.4 Hz, 1H), 2.65–2.59 (m, 2H, CH_2_), 2.54–2.53 (m, 1H), 1.65–1.54 (m, 2H, CH_2_). ^13^C NMR (151 MHz, DMSO-*d*_6_) *δ* 157.8, 150.9, 149.5, 136.8, 130.0, 126.4, 125.6, 120.7, 119.5, 113.2, 101.8, 94.3, 91.4, 75.8, 75.3, 65.6, 61.6, 59.4, 57.3, 55.0, 52.7, 52.2, 47.8, 32.7. HRMS (ESI): *m*/*z* calcd for C_27_H_31_N_7_O_2_ [M+H]^+^: 486.2612; found 486.2614.

2-((4-(cyclopropanecarbonyl)piperazin-1-yl)methyl)-5-(1*H*-indol-4-yl)-7-(morpholin-4-yl)pyrazolo[1,5-*a*]pyrimidine (**44**)

Compound **44** was prepared from aldehyde **22** (0.12 g, 0.36 mmol), 1-(cyclopropylcarbonyl)piperazine (64 µL, 70 mg, 0.43 mmol), DCM (4.0 mL) and sodium triacetoxyborohydride (0.11 g, 0.54 mmol). The crude product was purified by flash chromatography (0–10% MeOH gradient in AcOEt) to give **44** (0.14 g, 0.29 mmol) as a light yellow solid with 78% yield. ^1^H NMR (500 MHz, CDCl_3_) *δ* 8.67 (s, 1H, NH), 7.60 (d, *J* = 7.3 Hz, 1H, Ar-H), 7.48 (d, *J* = 8.1 Hz, 1H, Ar-H), 7.33–7.27 (m, 2H, Ar-H), 7.10 (s, 1H, Ar-H), 6.65 (s, 1H, Ar-H), 6.64 (s, 1H, Ar-H), 4.01–3.95 (m, 4H, morph.), 3.84 (s, 2H, CH_2_), 3.80–3.75 (m, 4H, morph.), 3.75–3.65 (m, 4H), 2.69–2.55 (m, 4H), 1.75–1.69 (m, 1H, CH), 1.01–0.95 (m, 2H, CH_2_), 0.77–0.72 (m, 2H, CH_2_). ^13^C NMR (126 MHz, CDCl_3_) *δ* 174.81, 153.02, 139.60, 128.89, 128.34, 124.80, 123.14, 115.62, 105.57, 98.80, 95.28, 69.17, 59.40, 51.37, 13.80, 10.22. HRMS (ESI): *m*/*z* calcd for C_27_H_31_N_7_O_2_ [M+H]^+^: 486.2612; found 486.2619.

2-((4-(cyclopropylmethyl)piperazin-1-yl)methyl)-5-(1*H*-indol-4-yl)-7-(morpholin-4-yl)pyrazolo[1,5-*a*]pyrimidine (**45**)

Compound **45** was prepared from aldehyde **22** (85 mg, 0.25 mmol), 1-(cyclopropylmethyl)piperazine (44 µL, 41 mg, 0.29 mmol), DCM (3.0 mL) and sodium triacetoxyborohydride (78 mg, 0.37 mmol). The crude product was purified by flash chromatography (0–20% MeOH gradient in AcOEt) to give **45** (97 mg, 0.21 mmol) as a light yellow solid with 84% yield. ^1^H NMR (400 MHz, DMSO-*d*_6_) *δ* 11.34 (s, 1H, NH), 7.65–7.63 (m, 1H, Ar-H), 7.53 (d, *J* = 8.1 Hz, 1H, Ar-H), 7.47–7.46 (m, 1H, Ar-H), 7.21 (t*, J* = 7.7 Hz, 1H, Ar-H), 7.10–7.09 (m, 1H, Ar-H), 6.76 (s, 1H, Ar-H), 6.48 (s, 1H, Ar-H), 3.86–3.84 (m, 4H, morph.), 3.78–3.77 (m, 4H, morph.), 3.65 (s, 2H, CH_2_), 2.53–2.45 (m, 8H, piperaz.), 2.15 (d, *J* = 6.6 Hz, 2H, CH_2_), 0.84–0.77 (m, 1H, CH), 0.45–0.40 (m, 2H, CH_2_), 0.06–0.02 (m, 2H, CH_2_). ^13^C NMR (101 MHz, DMSO-*d*_6_) *δ* 157.9, 154.0, 151.0, 149.5, 136.8, 130.0, 126.4, 125.6, 120.7, 119.5, 113.2, 101.8, 94.7, 91.5, 65.6, 62.8, 56.1, 52.7, 52.7, 47.9, 8.2, 3.7. HRMS (ESI): *m*/*z* calcd for C_27_H_33_N_7_O [M+H]^+^: 472.2819; found 472.2825.

Procedure for [5-chloro-7-(morpholin-4-yl)pyrazolo[1,5-a]pyrimidin-2-yl]methanol (**46**)

To the solution of compound **19** (16.6 g, 46.3 mmol) in CHCl_3_ (150 mL), methanesulfonic acid (61 mL, 925 mmol) was added, and then the reaction mixture was stirred at room temperature. After two h, the reaction mixture was poured onto the water containing ice and alkalized with 15% sodium hydroxide solution (25 mL). The aqueous phase was extracted with ethyl acetate (35 mL), and after separation, the organic phase was dried over anhydrous sodium sulfate. After filtration of the drying agent and evaporation of the solvent, the residue was purified by column chromatography (0–80% ethyl acetate gradient in heptane) to give **46** (12 g, 44.76 mmol) with 97% yield as an off-white solid. ^1^H NMR (300 MHz, CDCl_3_) *δ*: 6.49 (s, 1H, Ar-H), 6.07 (s, 1H, Ar-H), 4.87 (s, 2H, CH_2_), 4.00–3.90 (m, 4H, morph.), 3.83–3.73 (m, 4H, morph.). MS-ESI: *m*/*z* calcd for C_11_H_13_ClN_4_O_2_ [M+H]^+^: 269.08; found 269.0.

Procedure for 5-chloro-7-(morpholin-4-yl)pyrazolo[1,5-a]pyrimidine-2-carbaldehyde (**47**)

To a solution of compound **46** (3.00 g, 10.9 mmol) in DMF (30.0 mL) in argon atmosphere was added Dess–Martin periodinane (97%, 5.74 g, 13.1 mmol). The resulting mixture was stirred at room temperature for 2 h. The solvent was evaporated. The residue was washed with AcOEt and filtered. The filtrate was concentrated, and the crude product was purified by flash chromatography (0–100% AcOEt gradient in heptane) to give **47** (1.34 g, 5.02 mmol) with 46% yield. ^1^H NMR (500 MHz, DMSO-*d*_6_) δ 10.09 (s, 1H, CHO), 6.97 (s, 1H, Ar-H), 6.63 (s, 1H, Ar-H), 3.90–3.85 (m, 4H, morph.), 3.86–3.78 (m, 4H, morph.).

Procedure for 2-(1-{[5-chloro-7-(morpholin-4-yl)pyrazolo[1,5-a]pyrimidin-2-yl]methyl}piperidin-4-yl)propan-2-ol (**48**)

To the solution of compound **47** (3.4 g, 12.5 mmol) in dry DCM (30 mL), 2-(4-piperidyl)-2-propanol (2.24 g, 15.0 mmol) was added and then stirred at room temperature. After one hour, sodium triacetoxyborohydride (4.59 g, 21.2 mmol) was added, stirring the mixture at room temperature for a further 15 h. Then, water (45 mL) was added to the reaction mixture, and water-organic phases were separated. The aqueous phase was extracted three times with DCM (30 mL). Combined organic phases were dried over anhydrous sodium sulfate, filtered, and evaporated under reduced pressure. The residue was purified by flash chromatography (0–10% methanol gradient in ethyl acetate) to give **48** (3.1 g, 7.88 mmol) with a 63% yield as a slightly yellow solid. ^1^H NMR (500 MHz, CDCl_3_) *δ* 6.47 (s, 1H, Ar-H), 6.01 (s, 1H, Ar-H), 3.96–3.91 (m, 4H, morph.), 3.81–3.76 (m, 4H, morph.), 3.71 (s, 2H), 3.11–3.00 (m, 2H), 2.09–1.98 (m, 2H), 1.78–1.67 (m, 2H), 1.48–1.35 (m, 2H), 1.30–1.23 (m, 1H), 1.17 (s, 6H, 2xCH_3_). MS-ESI: *m*/*z* calcd for C_19_H_28_ClN_5_O_2_ [M+H]^+^: 394.20; found 394.1.

Procedure for 2-((4-(2-hydroxypropan-2-yl)piperidin-1-yl)methyl)-5-(5-fluoro-1H-indol-4-yl)-7-(morpholin-4-yl)pyrazolo[1,5-a]pyrimidine (**49**)

Compound **49** was prepared according to the general procedure for the Suzuki reaction. Synthesized from **48** (0.15 g, 0.381 mmol), 5-fluoro-4-(4,4,5,5-tetramethyl-1,3,2-dioxaborolan-2-yl)-1*H*-indole (0.16 g, 0.571 mmol), tetrakis(triphenylphosphine)palladium(0) (90 mg, 0.076 mmol), 2M aqueous sodium carbonate solution (0.38 mL, 0.762 mmol) and DME (6 mL). The crude product was purified by flash chromatography (50–100% ethyl acetate gradient in heptane) to give **49** (0.11 g, 0.22 mmol) with 60% yield. ^1^H NMR (400 MHz, DMSO-*d*_6_) *δ* 11.34 (s, 1H, NH), 7.50–7.46 (m, 2H, Ar-H), 7.07–7.02 (m, 1H, Ar-H), 6.72–6.71 (m, 1H, Ar-H), 6.56 (d, *J* = 1.7 Hz, 1H, Ar-H), 6.49 (s, 1H), 4.02 (bs, 1H), 3.84–3.80 (m, 4H, morph.), 3.77–3.73 (m, 4H, morph.), 3.63 (s, 2H), 2.98–2.95 (m, 2H), 1.95–1.90 (m, 2H), 1.65–1.62 (m, 2H), 1.31–1.23 (m, 3H), 1.01 (s, 6H, 2xCH_3_). ^13^C NMR (101 MHz, DMSO-*d*_6_) *δ* 154.5, 154.2, 153.6, 150.8, 149.2, 132.8, 128.0, 126.9, 116.6, 113.5, 109.6, 101.8, 94.8, 93.9, 70.2, 65.6, 56.4, 53.9, 47.8, 46.9, 26.9, 26.6. HRMS (ESI): *m*/*z* calcd for C_27_H_33_FN_6_O_2_ [M+H]^+^: 493.2722; found 493.724.

Procedure for 2-((4-(2-hydroxypropan-2-yl)piperidin-1-yl)methyl)-5-(1H-pyrrolo [2,3-c]pyridin-4-yl)-7-(morpholin-4-yl)pyrazolo[1,5-a]pyrimidine (**50**)

Compound **50** was prepared according to the general procedure for the Suzuki reaction. Synthesized from **48** (0.15 g, 0.381 mmol), 6-azaindole-4-boronic acid pinacol ester (0.15 g, 0.571 mmol), tetrakis(triphenylphosphine)palladium(0) (88 mg, 0.076 mmol), 2M aqueous sodium carbonate solution (0.38 mL, 0.762 mmol) and DME (6 mL). The crude product was purified by flash chromatography (0–2% methanol gradient in ethyl acetate) to give **50** (0.13 g, 0.27 mmol) with 72% yield. ^1^H NMR(300 MHz, CDCl_3_) *δ* 10.43 (bs; 1H, NH); 8.80–8.78 (m; 1H, Ar-H); 8.72 (s; 1H, Ar-H); 7.51 (d; *J* = 3.1 Hz; 1H, Ar-H); 7.18 (d; *J* = 2.6 Hz; 1H, Ar-H); 6.65 (s; 1H, Ar-H); 6.61 (s; 1H, Ar-H); 4.02–3.90 (m; 4H, morph.); 3.84–3.72 (m; 6H); 3.20–3.09 (m; 2H, CH_2_); 2.16–2.03 (m; 2H, CH_2_); 1.80–1.70 (m; 2H); 1.53–1.31 (m; 3H); 1.18 (s; 6H, 2xCH_3_). ^13^C NMR (75 MHz, CDCl_3_) *δ* 156.4, 154.9, 151.7, 150.5, 138.2, 135.1, 133.8, 131.2, 130.0, 126.6, 102.7, 96.3, 91.5, 72.6, 66.4, 57.0, 54.3, 48.6, 47.4, 27.1, 27.1. HRMS (ESI): *m*/*z* calcd for C_26_H_33_N_7_O_2_ [M+H]^+^: 476.2769; found 476.2775.

Procedure for 2-((4-(2-hydroxypropan-2-yl)piperidin-1-yl)methyl)-5-(1H-pyrrolo [2,3-b]pyridin-4-yl)-7-(morpholin-4-yl)pyrazolo[1,5-a]pyrimidine (**51**)

Compound **51** was prepared according to the general procedure for the Suzuki reaction. Synthesized from **48** (0.15 g, 0.381 mmol), 4-(4,4,5,5-tetramethyl-1,3,2-dioxaborolan-2-yl)-1*H*-pyrrolo [2,3-*b*]pyridine (0.14 g, 0.571 mmol), tetrakis(triphenylphosphine) palladium(0) (88 mg, 0.076 mmol), 2M aqueous sodium carbonate solution (0.38 mL, 0.762 mmol) and DME (6 mL). The crude product was purified by flash chromatography (0–5% methanol gradient in ethyl acetate) to give **51** (0.12 g, 0.25 mmol) with 67% yield. ^1^H NMR (400 MHz, DMSO-*d*_6_) *δ* 11.85 (s, 1H, NH), 8.34 (d, *J* = 5.0 Hz, 1H, Ar-H), 7.67 (d, *J* = 5.0 Hz, 1H, Ar-H), 7.61–7.59 (m, 1H, Ar-H), 7.10–7.09 (m, 1H, Ar-H), 6.84 (s, 1H, Ar-H), 6.55 (s, 1H, Ar-H), 4.02 (s, 1H), 3.84–3.83 (m, 8H), 3.63 (s, 2H), 2.97–2.94 (m, 2H), 1.95–1.89 (m, 2H), 1.65–1.62 (m, 2H), 1.27–1.20 (m, 3H), 1.01 (s, 6H, 2xCH_3_). ^13^C NMR (101 MHz, DMSO-*d*_6_) *δ* 155.5, 154.8, 150.9, 149.8, 149.7, 142.5, 137.0, 127.4, 117.1, 114.1, 100.8, 95.2, 91.1, 70.2, 65.6, 56.4, 53.9, 47.9, 46.8, 26.9, 26.6.HRMS (ESI): *m*/*z* calcd for C_26_H_33_N_7_O_2_ [M+H]^+^: 476.2769; found 476.2776.

Procedure for 4-{2-[(4-tert-butylpiperazin-1-yl)methyl]-5-chloropyrazolo[1,5-a]pyrimidin-7-yl}morpholine (**52**)

To the solution of compound **47** (4.1 g, 15.4 mmol) in dry DCM (60 mL), *N*-*t*-butylpiperazine (2.62 g, 18.4 mmol) was added and then stirred at room temperature. After one h, sodium triacetoxyborohydride (5.54 g, 26.1 mmol) was added, and the mixture was stirred at room temperature for a further 15 h. Then, water (50 mL) was added to the reaction mixture, and the phases were separated. The aqueous phase was extracted three times with DCM (45 mL). Combined organic phases were dried over anhydrous sodium sulfate, filtered, and evaporated under reduced pressure. The residue was purified by flash chromatography (0–10% methanol gradient in ethyl acetate) to give **52** (3.2 g, 8.15 mmol) with 53% yield as a slightly yellow solid. ^1^H NMR (300 MHz, CDCl_3_) *δ* 6.47 (s, 1H, Ar-H), 6.03 (s, 1H, Ar-H), 3.99–3.89 (m, 4H, morph.), 3.84–3.76 (m, 4H, morph.), 3.74 (s, 2H, CH_2_), 2.63 (s, 8H, piperaz.), 1.08 (s, 9H, *t*-Bu.). MS-ESI: *m*/*z* calcd for C_19_H_29_ClN_6_O [M+H]^+^: 393.22; found 393.1.

Procedure for 2-((4-tert-butylpiperazin-1-yl)methyl)-5-(5-fluoro-1H-indol-4-yl)-7-(morpholin-4-yl)pyrazolo[1,5-a]pyrimidine (**53**)

Compound **53** was prepared according to the general procedure for the Suzuki reaction. Synthesized from **52** (0.12 g, 0.305 mmol), 5-fluoro-4-(4,4,5,5-tetramethyl-1,3,2-dioxaborolan-2-yl)-1*H*-indole (0.13 g, 0.458 mmol), tetrakis(triphenylphosphine)palladium(0) (72 mg, 0.061 mmol), 2M aqueous sodium carbonate solution (0.31 mL, 0.611 mmol) and DME (5 mL). The crude product was purified by flash chromatography (0–5% methanol gradient in ethyl acetate) to give **53** (0.10 g, 0.20 mmol) with 68% yield. ^1^H NMR (600 MHz, DMSO-*d*_6_) *δ* 11.36 (s, 1H, NH), 7.50–7.48 (m, 1H, Ar-H), 7.47–7.46 (m, 1H, Ar-H), 7.06–7.03 (m, 1H, Ar-H), 6.71–6.70 (m, 1H, Ar-H), 6.57–6.57 (m, 1H, Ar-H), 6.51 (s, 1H, Ar-H), 3.83–3.82 (m, 4H, morph.), 3.75–3.74 (m, 4H, morph.), 3.67 (s, 2H, CH_2_), 2.65–2.37 (m, 8H, piperaz.), 1.03 (s, 9H, *t*-Bu.). ^13^C NMR (151 MHz, DMSO-*d*_6_) *δ* 154.9, 153.7, 153.4, 150.8, 149.2, 132.8, 128.0, 126.9, 116.6, 113.6, 109.7, 101.8, 95.0, 94.0, 65.6, 55.8, 53.3, 47.8, 45.3, 40.0, 25.5. HRMS (ESI): *m*/*z* calcd for C_27_H_34_FN_7_O [M+H]^+^: 492.2881; found 492.2886.

Procedure for 2-((4-tert-butylpiperazin-1-yl)methyl)-5-(1H-pyrrolo [2,3-c]pyridin-4-yl)-7-(morpholin-4-yl)pyrazolo[1,5-a]pyrimidine (**54**)

Compound **54** was prepared according to the general procedure for the Suzuki reaction. Synthesized from **52** (0.12 g, 0.305 mmol), 6-azaindole-4-boronic acid pinacol ester (0.12 g, 0.458 mmol), tetrakis(triphenylphosphine)palladium(0) (71 mg, 0.061 mmol), 2M aqueous sodium carbonate solution (0.305 mL, 0.611 mmol) and DME (5 mL). The crude product was purified by flash chromatography (0–5% methanol gradient in ethyl acetate) to give **54** (0.11 g, 0.23 mmol) with 77% yield. ^1^H NMR (600 MHz, DMSO-*d*_6_) *δ* 11.84 (s, 1H, NH), 8.83–8.83 (m, 1H, Ar-H), 8.76–8.75 (m, 1H, Ar-H), 7.73–7.73 (m, 1H, Ar-H), 7.17–7.16 (m, 1H, Ar-H), 6.84 (s, 1H, Ar-H), 6.51 (s, 1H, Ar-H), 3.85–3.84 (m, 5H), 3.82–3.81 (m, 4H), 3.63 (s, 2H, CH_2_), 2.54–2.46 (m, 8H, piperaz.), 0.97 (s, 9H, *t*-Bu.). ^13^C NMR (151 MHz, DMSO-*d*_6_) *δ* 155.8, 154.1, 151.0, 149.6, 137.9, 133.6, 130.6, 129.8, 125.1, 101.7, 94.9, 90.9, 65.6, 56.0, 53.6, 52.9, 47.9, 45.2, 40.0, 25.7. HRMS (ESI): *m*/*z* calcd for C_26_H_34_N_8_O [M+H]^+^: 475.2928; found 472.2929.

Procedure for 2-((4-tert-butylpiperazin-1-yl)methyl)-5-(1H-pyrrolo [2,3-b]pyridin-4-yl)-7-(morpholin-4-yl)pyrazolo[1,5-a]pyrimidine (**55**)

Compound **55** prepared according to the general procedure for the Suzuki reaction. Synthesized from **52** (0.15 g, 0.382 mmol), 4-(4,4,5,5-tetramethyl-1,3,2-dioxaborolan-2-yl)-1*H*-pyrrolo [2,3-*b*]pyridine (0.14 g, 0.573 mmol), tetrakis(triphenylphosphine) palladium(0) (88 mg, 0.076 mmol), 2M aqueous sodium carbonate solution (0.38 mL, 0.763 mmol) and DME (5 mL). The crude product was purified by flash chromatography (0–5% methanol gradient in ethyl acetate) to give **55** (0.13 g, 0.27 mmol) with 72% yield. ^1^H NMR (400 MHz, DMSO-*d*_6_) *δ* 11.85 (s, 1H, NH), 8.34 (d, *J* = 5.0 Hz, 1H, Ar-H), 7.67 (d, *J* = 5.1 Hz, 1H, Ar-H), 7.61–7.59 (m, 1H, Ar-H), 7.10–7.08 (m, 1H, Ar-H), 6.85 (s, 1H, Ar-H), 6.56 (s, 1H, Ar-H), 3.84–3.83 (m, 8H, morph.), 3.64 (s, 2H, CH_2_), 2.50–2.43 (m, 8H, piperaz.), 0.97 (s, 9H, *t*-Bu.). ^13^C NMR (101 MHz, DMSO-*d*_6_) *δ* 155.5, 154.4, 150.9, 149.8, 142.5, 137.0, 127.4, 117.1, 114.1, 100.8, 95.3, 91.2, 65.6, 55.9, 53.5, 53.1, 47.9, 45.2, 40.2, 25.7. HRMS (ESI): *m*/*z* calcd for C_26_H_34_N_8_O [M+H]^+^: 475.2928; found 472.2936.

### 3.2. Docking Study

The docking procedure was performed in the PI3K *δ* protein from Protein Data Bank (PDB: 2WXP) using the Auto-Dock Vina program [55]. All figures with examples of 3D modeling of a possible binding mode of selected compounds were prepared based on the calculated pK_a_ from the Instant JChem 21.13.0 program [57]. More specifically, all structures depicted in the respective figures have not had protons added, but the appropriate protonation state has been maintained.

### 3.3. Biology

#### 3.3.1. In Vitro Kinase Inhibition Assay for PI3K

Tested compounds were dissolved in 100% DMSO, and obtained solutions were serially diluted in 1× reaction buffer. The recombinant kinase solution was diluted in a reaction mixture comprising 5× reaction buffer, respective compound solution (1 mM sodium diacetate 4,5-bisphosphate phosphatidylinositol (PIP2) solution in 40 mM Tris buffer), and water. In a 96-wells plate, 5 μL of compound solutions and 15 μL of the kinase solution in the reaction mixture were added per well. To initiate the interaction of chemical compounds to be tested with the enzyme, the plate was incubated for 10 min at a suitable temperature in a plate thermostat with orbital shaking at 600 rpm. Negative control wells contained all the above reagents except tested compound and kinase, and positive control wells contained all the above reagents except tested compounds. The enzymatic reaction was initiated by adding 5 μL of 150 μM ATP solution. Subsequently, the plate was incubated for 1 h at 25 or 30 °C (depending on the PI3K isoform tested) in a plate thermostat with orbital shaking of the plate contents at 600 rpm. The reaction conditions are combined in the table below (Table 6).

Detection of ADP formed in the enzymatic reaction was then performed using ADP-Glo Kinase Assay™ (Promega, Madison, WI, USA). To the wells of a 96-well plate, 25 μL of ADP-Glo Reagent™ was added, and the plate was incubated for 40 min at 25 °C in a plate thermostat with orbital shaking at 600 rpm. Then 50 μL of Kinase Detection Reagent were added to each well, and the plate was incubated for 40 min at 25 °C in a plate thermostat with orbital shaking at 600 rpm. Once the incubation was complete, the luminescence intensity was measured using a Victor Light luminometer (Perkin Elmer, Inc., Waltham, MA, USA). IC_50_ values were determined based on luminescence intensity measured in wells containing tested compounds at different concentrations in relation to control wells. These values were calculated with Graph Pad 5.03 software by fitting the curve using non-linear regression. Each compound was tested at least in quadruplicates (4 wells) on two 96-well plates utilizing at least 4 wells for each control. Averaged results of inhibition activity respective to specific isoforms of PI3K kinases for tested compounds are presented as IC_50_ values in Table 1, Table 2, Table 3 and Table 4.

#### 3.3.2. Influence of Selected Compounds on B Cells Proliferation

CD19 cells were isolated from PBMC using magnetic beads (Stem Cell, Cambridge, MA, USA) and then labeled with 2 µM CFSE (Invitrogen, Waltham, MA, USA).

1 × 10^5^ cells were seeded on 96-well plate, activated by 2 µg/mL *α*IgM (Jackson ImmunoResearch, Ely, UK) and 1 µg/mL ODN2006 (InvivoGen, San Diego, CA, USA), and incubated with increasing concentrations of drugs (0.1, 0.3, 1.0, 3.3, 10, 33, 100, 333, 1000, 3333, 10,000 nM). After four days, cells were stained with LIVE/DEAD™ kit (Invitrogen, Waltham, MA, USA). Samples were acquired using Attune NxT Flow Cytometer (Invitrogen, Waltham, MA, USA) and analyzed using FlowJo software. Each biological assay was performed with cells isolated from a different donor. The presented results constitute the average value of the percentage of proliferating cells from 3 independent experiments.

### 3.4. Metabolic Stability and Solubility

#### 3.4.1. Metabolic Stability Assay

Assessment of metabolic phase I stability in mouse (CD-1™) and human microsomes (Thermo-Fisher Scientific, Waltham, MA, USA) was performed on 96-well non-binding plates (Greiner, Frickenhausen, Germany) at 1 μM concentration for verapamil (positive control) and donepezil (negative control) and tested compounds. Unless otherwise stated, all chemicals and materials were ordered from Merck Life Science (Palo Alto, CA, USA). Each biological replicate was prepared in triplicates. Briefly, mixtures were incubated in 100 mM potassium phosphate buffer with microsomes (0.5 mg/mL) and NADPH (1–1.2 mM) on a plate shaker (500 rpm) in the dark at 37 °C. A 4× solution of NADPH, a cofactor for metabolic enzymes, was prepared directly prior to the experiment by reducing NADP with G6P dehydrogenase (13.2 mM MgCl_2_, 13.2 mM G6P, 5.2 mM NADP, 3.2 U/mL G6P dehydrogenase, 20 min at 30 °C, 500 rpm). The negative control contained buffer instead of NADPH solution. Samples were collected at 0, 10, 20, and 40 min or 0 and 40 min for the negative and double negative controls. The reaction was stopped by protein precipitation in 2 volumes of ice-cold MeOH with 200 nM imipramine (an internal standard for LC-MS analysis). Then, the extract was mixed (1 min, 1000 rpm), filtered through a 0.22 μm filter on a 96-well plate vacuum manifold, and subjected to LC-MS analysis.

#### 3.4.2. Kinetic Stability Assay

Kinetic solubility was determined by the shake-flask protocol [58,59]. Appropriate compounds (500 µM) were incubated in an aqueous buffer (0.1 M phosphate-buffered saline pH 7.4) at 25 °C with stirring at 500 rpm. The samples were taken at the start time and after 24 h of incubation, filtered through 0.22 µm filters, and diluted with two volumes of acetonitrile. UHPLC-UV/Vis determined sample concentrations. A calibration curve was prepared to quality the compound’s contents in the test solution.

## 4. Conclusions

Based on the 2-methyl-pyrazolo[1,5-*a*]pyrimidine system, the most promising R^1^ (Figure 1) substituent in terms of activity and selectivity was selected, and appropriate structures were designed and synthesized in multi-step synthesis. Among various derivatives obtained, two amino groups were identified as the most promising concerning the PI3K*δ* activity and other PI3K isoforms selectivity: 2-(piperidin-4-yl) propan-2-ol and *N*-*tert*-butylpiperazine located at the C(2) position of the pyrazolo[1,5-*a*]pyrimidine. The most selective compounds turned out to be 4-{2-[(4-*tert*-butylpiperazin-1-yl)methyl]-7-(morpholin-4-yl)pyrazolo[1,5-*a*]pyrimidin-5-yl}-1*H*-indole (**37**) and 4-{2-[(4-*tert*-butylpiperazin-1-yl)methyl]-5-{1*H*-pyrrolo [2,3-*c*]pyridin-4-yl}pyrazolo[1,5-*a*]pyrimidin-7-yl}morpholine (**54**), bearing the indol or azaindole system as the R^1^ substituent and *N*-*tert*-butylpiperazine as the R^2^ (Figure 2) residue. Molecular calculations and docking studies supported the strong tryptophan shelf (Trp-760) mechanism in which the lipophilic *tert*-butyl substituent is possibly engaged. Compound **54** (CPL302253) showed promising additional properties such as suitable kinetic solubility or higher metabolic stability (Table 6) compared to compound **37**. For these reasons, CPL302253 was selected as a promising clinical candidate for the treatment of asthma. Additional, biological studies supporting this selection have been published by Gunerka et al. [15].

## Data Availability

Data is contained within the article and Appendix A.

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
