# Peer review of "Design, Synthesis, and Development of pyrazolo[1,5-a]pyrimidine Derivatives as a Novel Series of Selective PI3Kδ Inhibitors: Part I—Indole Derivatives"

_pharmaceuticals, 2022, doi:10.3390/ph15080949_

Round 1

Reviewer 1 Report

This submission described the development of pyrazolo-pyrimidine derivatives as PI3K-delta inhibitors. The work flow followed the order of virtual screening and evaluation on the enzymatic inhibition. The authors should improve the design concept and conduct some further experiments. I recommend a major revision.

1. What did “Part I – Indole derivatives” mean? Were there Part II or Part III?

2. The docking simulation was based on the complex containing GDC-0941, which is a pan-blocker. It was confusing that the authors designed a selective inhibitor for delta isoform based on the binding pattern of a pan-blocker. The design concept should be seriously checked and revised to construct a rational process.

3. The backbone structure also suggested that the synthesized series in this work was quite different from the selective inhibitors CDZ 173 and UCB-5857. Please include their features into the revised design concept.

4. The drawing of the structures should be standard. ACS 1996 pattern is recommended.

5. The authors should find the balance between the potency and the selectivity. In particular, in Table 4, the final choice, I think there was no obvious difference between 37 and 54. The authors should conduct some further experiments in models to check their real performance.

6. How about the toxicity?

7. In the docking simulation, the key residues were different from the reported ones. Please check it.

8. If possible, the authors should check the up-stream or down-stream nodes in the mechanism network.

9. The language use should be improved.

Reviewer 2 Report

The manuscript entitled “Design, synthesis, and development of pyrazolo[1,5-a]pyrimidine derivatives as a novel series of selective PI3Kδ inhibitors. Part I – Indole derivatives” is a wonderful study but I have one notice

The 1HNMR and 13CNMR are written in a general manner, it will be more convenient to be written in a specific manner. Which value for which proton or carbon.

Round 2

Reviewer 1 Report

The authors have dealt with all the comments rationally. I recommend the acceptance of the submission. A minor point: the authors should check the language use.